



# Seasonal contrast in size distributions and mixing state of black carbon and its association with PM1.0 chemical composition from the eastern coast of India

Sobhan Kumar Kompalli[1], Surendran Nair Suresh Babu[1], Sreedharan Krishnakumari Satheesh[2,3],

5   Krishnaswamy Krishna Moorthy[2], Trupti Das[4], Ramasamy Boopathy[4], Dantong Liu[5,6], Eoghan Darbyshire[5], James Allan[5,7], James Brooks[5], Michael Flynn[5], Hugh Coe[5]

[1]Space Physics Laboratory, Vikram Sarabhai Space Centre, India.

[2]Centre for Atmospheric & Oceanic Sciences, Indian Institute of Science, India.

10   [3]Divecha Centre for Climate Change, Indian Institute of Science, Bangalore, India

[4]Institute of Minerals and Materials Technology, CSIR, Bhubaneswar.

[5]Centre for Atmospheric Science, School of Earth and Environmental Sciences, University of Manchester, Manchester, UK.

[6]Department of Atmospheric Sciences, School of Earth Sciences, Zhejiang University, Hangzhou, Zhejiang, China.

[7]National Centre for Atmospheric Science, UK.

*Correspondence to*: S. Suresh Babu (sureshsplvssc@gmail.com), Sobhan K. Kompalli (sobhanspl@gmail.com)



## Abstract

Over the Indian region, aerosol absorption is considered to have potential impact on regional climate, monsoon and hydrological cycle. Black carbon (BC) is the dominant absorbing aerosol, whose absorption potential is largely determined by its microphysical properties, including its concentration, size and mixing state with other aerosol components. The Indo-Gangetic Plains (IGP) is one of the regional aerosol hot spots with diverse sources, both natural and anthropogenic, but still the information on the mixing state of the IGP aerosols, especially BC, is limited and a major source of uncertainty in understanding their climatic implications. In this context, we present the results from intensive measurements of refractory BC (rBC) carried out over Bhubaneswar, an urban site in the eastern coast of India, which experiences contrasting airmasses (the IGP outflow or coastal/marine airmasses) in different seasons.  This study helps to elucidate the microphysical characteristics of BC over this region and delineates the IGP outflow from the other airmasses. The observations were carried out as part of "South West Asian Aerosol Monsoon Interactions (SWAAMI)" collaborative field experiment during July 2016-May 2017, using a single particle soot photometer (SP2) that uses a laser-induced incandescence technique to measure the mass and mixing state of individual BC particles and an aerosol chemical speciation monitor (ACSM). Results highlighted the distinctiveness in aerosol microphysical properties in the IGP airmasses.  BC mass concentration was highest during winter (~1935 ± 1578 ng m$^{-3}$), when the prevailing air masses were mostly of IGP origin, followed by post-monsoon (mean ~1338 ± 1396 ng m$^{-3}$).  Mass median diameter (MMD) of the BC mass size distributions were in the range 0.190-0.195 µm suggesting mixed sources of BC, and further, higher values (~1.3-1.8) of bulk relative coating thickness (RCT) (ratio of optical and core diameters) were seen indicating a large fraction of highly coated BC aerosols in the IGP outflow. During the pre-monsoon, when marine/coastal airmasses prevailed, BC mass concentration was lowest (~816 ± 835 ng m$^{-3}$) and larger BC cores (MMD > 0.210 µm) were seen suggesting distinct source processes, while RCT was ~1.2-1.3, which may translate into higher extent of absolute coating on BC cores which may have important regional climate implications. During the summer monsoon, BC size distributions were dominated by smaller cores (MMD ≤ 0.185 µm) with lowest coating, indicating fresher BC, likely from fossil fuel sources. A clear diurnal variation pattern of BC and RCT was noticed in all the seasons, and day time peak in RCT suggested enhanced coating on BC due to the condensable coating material originated from photochemistry. Examination of sub-micron aerosol chemical composition highlighted that the IGP outflow is dominated by organics (47-49%) and marine/coastal airmasses contained greater amounts of sulphate (41-47%), while ammonium and nitrate were seen in minor amounts with significant concentrations only during the IGP airmass periods.  The diurnal pattern of sulphate resembled that of the RCT of rBC particles, whereas organic mass showed a pattern similar to that of the rBC mass concentration. Though the pre-monsoon is sulphate dominated, the coating on BC showed a negative association with sulphate and same is true for organic mass during the post-monsoon, suggesting preferential coating and importance of source processes (and co-emitted species) on the mixing state of BC. This is the first experimental data on the mixing state of BC from a long time series over the Indian region, and includes new information on black carbon in the IGP





outflow region. This data helps in improving the understanding of regional BC microphysical characteristics and their climate implications.

**Keywords**: Refractory black carbon, size distribution, mixing state, Indo-Gangetic Plain outflow



## 1. Introduction

The state of mixing of aerosols, especially that of absorbing aerosols, remains poorly quantified, despite its important role in determining the regional and global radiative impacts of aerosols and aerosol-cloud interactions (Bond et al., 2013; Liu et al., 2013; IPCC 2013). The importance of the Southwest Asian region need not be over-emphasized in this context; where the two most-absorbing aerosol species, Black carbon (BC) from a wide variety of sources in the locale and dust co-exist along with a large spectrum of other aerosol species such as sulphates, nitrates, phosphates, volatile organic compounds (VOC), and secondary organic aerosols (SOA) (Lee et al., 2002; Shiraiwa et al., 2007; Moteki et al., 2007; Moffet and Prather, 2009; Zhang et al., 2015). The dust is emitted from both local arid regions as well as advected from far off regions such as the Middle East and Eastern Africa, and these different sources are known to have distinctly differing absorption properties (Moorthy et al. 2007). Large seasonal changes in synoptic meteorology occur in this region throughout the year associated with the Asian monsoon system and the associated changes in atmospheric humidity and thermal convection, which is modulated by the local (meso-scale) meteorology and the regional orography. This interplay confines the aerosol between the Bihar Plateau to the South and the high Himalayan ranges to the north. Such strong variation is therefore likely to lead to significant changes in the aerosol characteristics, especially the state of mixing (Lawrence and Lelieveld 2010; Srivastava and Ramachandran, 2013; Srinivas and Sarin 2014; Moorthy et al., 2016; Raatikainen et al., 2017).

Both BC and dust are highly porous in nature (Adachi et al., 2010, 2014; China et al., 2012; Bond et al., 2013; Scarnato et al., 2015), and provide surface area for adhesion of other particulate and gaseous species, paving the way for surface based chemical reactions. In this regard, BC has some special significance. Nascent BC is hydrophobic, and is comprised of chain agglomerates with diameters of the order of few tens of nanometers. However, it collapses to spherical BC cores coated with other components via coagulation among aggregates and/or condensation of atmospheric vapours while aging in the atmosphere (Weingartner et al., 1997; Zuberi et al., 2005). Coatings of non-absorbing components on the core BC, alter the morphology of the BC and enhance the absorption potential of the resultant mixed phase particle to varying magnitudes through so called 'lensing effect' (e.g. Shiraiwa et al., 2010; Cappa et al., 2012; Peng et al., 2016; Ueda et al., 2016; Wang et al., 2016). In addition, the coating of other soluble species on BC affects its hygroscopic properties (Weingartner et al., 1997; McMeeking et al., 2011; Liu et al., 2013; Laborde et al., 2013), atmospheric life time and makes it more cloud condensation nuclei (CCN) active (Liu et al., 2013; IPCC 2013).

These have implications for direct and indirect radiative forcing of BC. Due to the rapid growth in the human population across the region and the large increase in vehicle use, large scale industrialization and increased anthropogenic activities in the last few decades, the overall aerosol burden over the Indian region has increased significantly and displays an increasing trend (Babu et al., 2013; Moorthy 2016). The Indo-Gangetic Plain (IGP) region is one of the aerosol hot spots with potential implications to regional radiative forcing (Nair et al., 2017) and circulation (Lawrence and Lelieveld 2010; Gautam et al., 2009) and has attracted wide attention. Large heterogeneity in the nature of sources over the IGP (industrial and vehicular emissions, crop residue and residential fuel burning) results in BC particles with varying microphysical properties (size,





concentrations and mixing state) which determine its absorption potential and radiative effects (Jacobson 2001; Cappa et al., 2012; Petzold et al., 2013; Bond et al., 2013). Hence it is essential to garner information on BC microphysical properties including its mixing state to understand its effect on absorption enhancement and further BC climatic implications (both direct and indirect effects).

5       Detailed characterization of the state of mixing of aerosols over the Indo-Gangetic Plains (IGP), which is recognized as one of the most complex regions as far as aerosols are concerned (Moorthy et al. 2016), remains virtually non-existent. There have been a few limited and isolated studies (Thamban et al., 2017 and references therein) that have been mainly based on chemical composition and theoretical/model calculations (Dey et al., 2008; Srivastava and Ramachandran, 2013) and did not explore BC mixing state due to inherent limitations of the methodologies employed. While characterization of BC

spectral absorption properties and its mass loading over India are numerous (e.g. Beegum et al., 2009; Kompalli et al., 2014; Prasad et al., 2018 and references therein), reports on the size distribution of BC and its mixing state are extremely limited and site/season specific (Raatikainen et al., 2017; Thamban et al., 2017). The non-availability of state-of-art instruments for near-real-time estimating of coating of BC core with other species has been one of the main reasons for such limited exploration. In this context, the single particle soot photometer, a laser induced incandescence technique, offers a powerful

tool for long-term measurements at single particle level (Moteki and Kondo, 2007; Schwarz et al., 2008, 2013; Laborde et al., 2012; Liu et al., 2014).

Recognizing the need for the above information for better understanding aerosol-radiation-cloud-monsoon interactions over the South Asian region, a super site (first of three) was established in July-2016 at Bhubaneswar located in the eastern IGP, as part of the joint Indo-UK experiment, "South West Asian Aerosol Monsoon Interactions (SWAAMI)" executed by the

Space Physics Laboratory of Indian Space Research Organisation (ISRO), the Indian Institute of Science (IISc), Bengaluru and the University of Manchester, United Kingdom. The key objectives have been (a) assessment of the impact of BC and co-emitted organic/inorganic species on the radiation budget via the direct, semi-direct and indirect effects, and (b) assessment of the impact of the aerosol radiative forcing on the local energy budget, atmospheric dynamics and hydrological cycle over India. Under this, state-of-the art instruments were installed at Bhubaneswar, which included a single particle soot

photometer (SP2) for characterization of refractory BC (rBC) aerosols based on long-term measurements, perhaps for the first time over the Indian region. This paper provides the details of the measurement and analysis, presents the results on the concentrations (mass and number), size distribution, mixing state in terms of coating thickness of BC, the seasonality and responses to contrasting airmass types. The contributions from distinct sources are examined and implications are discussed.

## 2. Experimental details

**2.1 Observational site, general meteorology and data period**





The measurements were carried out from Bhubaneswar, the capital city of Odisha state, in the eastern part of India (denoted by the star symbol in Figure1a), and is an aerial distance of 53 km from the western coastline of the Bay of Bengal. It is a moderately industrialized city, typically urbanized and with the consequent anthropogenic emissions (industrial, traffic and household). It receives outflow from the Indo-Gangetic Plain (IGP, indicated by oval symbol in the figure), lying to its west, through advection by the synoptic westerlies for most part of the year. The urban area is surrounded by rural regions along a radius of 70 km that host a variety of anthropogenic activities involving burning of solid fuel (wood, dung cake etc) for household cooking and small-scale industries such as brick kilns, and coal fired thermal power plants. It also comprises of a major shipping harbour (Mahapatra et al., 2013a; Venkatraman et al., 2005; Verma et al., 2012).

Aerosol measurements are carried out at this supersite from a custom-built container (Figure 1b) installed in the premises of the Institute of Minerals and Materials Technology (IMMT), Bhubaneswar (20.20°N, 85.80 °E, 78 amsl) which is away from the proximity of any major industrial and urban activities. Sampling of ambient aerosols is done by drawing air at 16.67 litres min$^{-1}$ from a height ~3 meter AGL through a stainless-steel tube fitted with a PM10 inlet. Another stainless tube, with inner diameter of ~0.635 cm and 30 cm length, was used for isokinetic subsampling from this main flow inside the stack. In order to keep the relative humidity of the sample flow < 50 %, a nafion membrane dryer was installed downstream the sample flow and further, using isokinetic flow splitters this flow was distributed among the various aerosol instruments. Data collection commenced from July 2016 and the data collected until May 2017 are used in this work.

Bhubaneswar experiences contrasting seasonal airmasses linked with the Asian monsoon system (Asnani, 1993). In Figure 2, isentropic five-day airmass back trajectories arriving at 100 meters AGL at the sampling site are shown, which were computed for all the individual days during: (a) summer monsoon (SMS) (June–September), (b) post-monsoon (PoMS) (October–November), (c) winter (December–February), (d) pre-monsoon (PMS) (March–May). It clearly reveals the dominance of the IGP outflow during the PoMS and winter season, while mixed (continental and marine/coastal) airmasses prevailed during the SMS, and predominantly coastal transit/marine air masses during the PMS. Thus, examination of the seasonal characteristics will help in delineating the distinctiveness of the IGP airmasses and characterising various source, sink and transformation (aging) processes. While contribution from fossil fuel sources will be less variant, the contribution from biomass burning sources varies seasonally. In supplementary Figure S1, spatial distribution of Moderate Resolution Imaging Spectroradiometer (MODIS) fire radiative power (MODIS Thermal Anomalies / Fire locations, Collection 6 product obtained from https://earthdata.nasa.gov/firms) for the representative months of different seasons; (a) August -2016 (SMS), (b) October -2016 (PoMS), (c) January -2017 (winter) and (d) May -2017 (PMS). Figure S1, clearly depicts seasonally varying source contributions with the highest amount of fire events occurring all over the Indian region during the PMS, whereas during the PoMS (northwest IGP) and winter (western, north eastern regions of India) less intense regional fire events are noticeable.

Table-1 depicts the seasonal average of several important meteorological parameters: temperature (T), relative humidity (RH), pressure (P) and wind speed (WS) along with maximum and minimum values recorded in that season measured using a collocated automatic weather station. During SMS, high RH (~ 77.1 ± 17.5 %), moderate temperatures (~



30.1 ± 3.8 °C) and moderate wind speeds (up to max. 5.2 m s$^{-1}$) prevailed. Also, this season witnessed heavy rainfall (total rainfall ~ 878 mm) associated with the monsoon. Compared to the SMS, lower temperature (~ 28.3 ± 6.3 °C), RH (~ 71.0 ± 21.2 %) and mean wind speeds (~1.07 ± 0.67 m s$^{-1}$) prevailed during the PoMS, with total rainfall (~ 201 mm) also being lower than during the SMS and temperature ranging from a minimum of 21 °C to a maximum of 34 °C. During winter, lower

temperatures (from a minimum of 18 °C to a maximum of 36 °C) and dry weather (mean RH 57.4 ± 22.9%) with calm wind conditions (mean wind speed ~ 0.95 ± 0.53 m s$^{-1}$) prevailed with almost nil rainfall, whereas the PMS witnessed the highest temperature of the year (mean ~33 ± 7.5 °C and range: 18.9 - 41 °C) with average wind speed ~1.97 ± 0.92 m s$^{-1}$ and a moderately humid atmosphere (mean RH ~ 67.1 ± 28.2 %). During this season the region received a total rainfall of ~149 mm associated with thunder shower events that led to high velocity local winds.

The sampling period spanned from 27-July, 2016 to 22-May, 2017 representing all the seasons of the year, atmospheric conditions and distinct prevailing airmasses. The data has been collected continuously, except for brief gaps during calibration, system checks (flow rate, chamber temperatures etc.) and preventive maintenance of the instruments or minor technical issues.

### 2.2 Instrumentation

In the present study data was collected using a single-particle soot photometer (SP2) (Make: Droplet Measurement Technologies, Boulder, USA) and an Aerosol Chemical Speciation Monitor (ACSM) (Make: Aerodyne Research Inc., USA). A scanning mobility particle sizer (SMPS) (to measure particle number size distributions) and a 7-channel Aethalometer were also operated, but they are not relevant to the present study.

The SP2 allows the characterization of the mixing state of refractive BC (rBC) of single particles by employing a

laser-induced incandescence technique and obtaining the scattering properties based on excitation by a 1064 nm Nd:YAG intracavity laser (Moteki and Kondo, 2007; Schwarz 2013, 2008; Laborde et al., 2012; Liu et al., 2014; Shiraiwa 2007). It also provides information about the number/mass concentrations and size distribution of rBC. While the amplitude of the scattering signal provides the information about the optical size ($D_p$) of the particle, the amplitude of the incandescence signal is proportional to the mass of the rBC. The mass equivalent diameter, or BC core diameter ($D_c$), is defined as the

diameter of a sphere containing the same mass of rBC as measured in the particle using a density, ρ ~1.8 g cm$^{-3}$ for atmospheric BC (Bond and Bergstrom, 2006; Moteki and Kondo, 2010; Moteki et al., 2010; McMeeking et al., 2011). Additionally, the scattering signal from the BC containing particles provides information about the scattering cross-section of the particle. However, since the particle is subjected to intense thermal heating and evaporation of the coating while passing through the laser beam, the scattering signal gets perturbed. This signal is reconstructed using the leading edge only (LEO)

fitting technique which uses the leading edge of the unperturbed scattering signal before its volatilization (Liu et al., 2014) from which the full scattering signal is reconstructed. The scattering enhancement can be determine by using the BC core size ($D_c$) to derive the optical diameter of the BC core of the particle and the coated BC size ($D_p$) and employing Mie


calculations where the whole particle is idealized as a two-component sphere with a concentric core-shell morphology. In the present study, we have used a core (rBC) refractive index value of 2.26 – 1.26i (Moteki et al., 2010; Liu et al., 2014; Taylor et al., 2014) and a coating refractive index of 1.5+0i, which is representative of the corresponding values determined for inorganic salts (e.g., ammonium sulphate) and secondary organic aerosol (Schnaiter et al., 2005; Lambe et al., 2013). To

quantify the extent of coating on the BC particle, relative coating thickness (RCT) and absolute coating thickness (ACT), defined as $D_p/D_c$ and $(D_p–D_c)/2$ respectively, were used. Liu et al., (2010, 2014) have described the configuration, operation and data interpretation procedures of this specific instrument in detail. Taylor et al. (2014) described the methodology to determine the $D_p/D_c$ in detail and examined the sensitivity of the derived parameters to the density and refractive index values. The SP2 was operated at a flow rate of 80 cm$^3$ min$^{-1}$ and was periodically calibrated. Aquadag® black carbon particle

standards (Aqueous Deflocculated Acheson Graphite, manufactured by Acheson Inc., USA) were used for the calibration of the SP2. However, as Aquadag®-generated particle standards do not represent ambient BC, a correction factor of 0.75 is incorporated (e.g., Moteki and Kondo 2010; Laborde et al., 2012).

Real time characterization of the non-refractory PM1.0 aerosol mass and composition was carried out using an ACSM (Ng et al., 2011). The ACSM was operated with 30 minutes sampling interval alongside the SP2. The ACSM uses

quadrupole mass spectrometry to chemically characterize the submicron (40-1000 nm range) particulate composition of the organic, sulphate, nitrate, ammonium and chloride components. Initially the particles are focused on to a resistively heated thermal vaporizer operating at 600 °C using an aerodynamic particle focusing lens in a high vacuum environment. The evaporated gas stream from the particle evaporation is detected after ionization via electron impact by the mass spectrometer. The mass spectra are used to extract chemical composition information. The differential pumping required for the separation

of gas from the particle beams is achieved by employing three turbo molecular pumps along with a main diaphragm pump. From a main flow rate of 3 litres min$^{-1}$, the ACSM draws a sample flow of 0.1 litres min$^{-1}$ using a 100 µm diameter critical aperture. The instrument is periodically calibrated and all the corrections described in Ng et al., (2011) were applied during the data post-processing. A real-time and composition dependent collection efficiency (CE) correction based on Middlebrook et al. (2012) was applied to account for the uncertainties arising due to the usage of standard vaporizer.

## 3.    Results and Discussion

### 3.1 Mass and number concentrations

Temporal variation of daily mean mass concentration of BC, and number concentrations of BC and non-BC scattering particles are shown respectively in Figure 3(a) and Figure 3(b). Number concentrations of BC and scattering particles peak in winter, while they are the lowest in PMS with moderate values through PoMS. The overall annual mean number

concentrations (and their standard deviation) of rBC and non-BC particles (that are in the detection range of the SP2 (200-





400 nm) only) are ~ 496 (± 536) cm$^{-3}$ and 702 (± 458 cm$^{-3}$) respectively, suggesting a large variability (and a skewed distribution) and the median values of rBC and non-BC particles are 333 cm$^{-3}$ and 595 cm$^{-3}$ respectively.

Similarly, the BC mass concentration also showed significant temporal variation with high values during winter and low values during PMS. During SMS the daily mean mass concentration ranged between 307-2581 ng m$^{-3}$ with a seasonal mean ~941 ± 615 ng m$^{-3}$ and during PoMS the seasonal mean was ~1338 ± 1396 ng m$^{-3}$ with daily mean values varying between 357-2869 ng m$^{-3}$. Winter witnessed enhanced BC mass concentrations with daily mean values ranging between 281-3680 ng m$^{-3}$ and seasonal mean of ~1935 ± 1578 ng m$^{-3}$, whereas the lowest concentrations were found during PMS with daily mean values varying between 52 - 3071 ng m$^{-3}$ and seasonal mean of ~816 ± 835 ng m$^{-3}$. Lower values during SMS may be attributed to wide spread precipitation across the region and decreased source strength during rainy periods. Conversely, the decrease in rainfall, prevailing calm wind conditions, coupled with decreased ventilation due to the shallow boundary layer as a result of prevailing lower temperatures, contributed to the build-up of aerosols during PoMS, which continued and was further enhanced during winter. During PMS, strong thermal convection resulting from enhanced solar heating of the dry land lifts the boundary layer to higher altitudes, and with winds gaining speed, there is greater dispersion of the aerosols (Kompalli et al., 2014) leading to substantial reduction in the surface concentrations. The annual mean rBC mass concentration is 1340 ± 1322 ng m$^{-3}$. In an earlier experiment during winter at Kanpur, a polluted urban area in the central IGP, Thamban et al., (2017) have reported rBC mass concentration in the range ~ 0.73 to 17.05 µg m$^{-3}$ with a mean (± standard deviation) ~4.06 ± 2.46 µg m$^{-3}$ which is about twice as high as we have seen at Bhubaneswar at the eastern fringe of the IGP. In a short experimental campaign during pre-monsoon season 2014, Raatikainen et al., (2017) have reported average rBC mass concentrations of 11 ± 11 µg m$^{-3}$ for Gual Pahari (an IGP site close to Delhi) and 1.0 ± 0.6 µg m$^{-3}$ for the high-altitude site Mukteshwar, in the foot hills of the central Himalayas. The significantly higher values seen over the central IGP stations are attributed to the proximity of local emissions. However, our values at Bhubaneswar are comparable to the mean rBC values reported by Liu et al., (2014) over London (~ 1.3 µg m$^{-3}$) and higher than the values over Paris city (~0.9 µg m$^{-3}$, Laborde et al., 2013); but are significantly lower than those reported over Chinese cities, Beijing (~5.5 µg m$^{-3}$, Wu et al., 2016), Shanghai (~ 3.2 µg m$^{-3}$, Gong et al., 2016), Shenzhen (~ 4.1 µg m$^{-3}$, Huang et al., 2012), Kaiping (~ 3.3 µg m$^{-3}$, Huang et al., 2011). Our values are higher than the values reported by Wang et al., (2018) over a remote site in the southern Tibetan Plateau (~ 0.31 ± 0.35 µg m$^{-3}$).

**3.2 Seasonal distinctiveness of BC size distribution and modal parameters**

The size distribution of BC cores is one of the important factors while determining light absorption characteristics of the aerosols and direct radiative forcing (Reddington et al., 2013). Knowing BC size distribution (in addition to its coating information) is also important to understand BC life cycle, as BC from different sources would have different sizes and different scavenging mechanisms display different size and composition dependent efficiencies which can effect BC aging process. In the present study, modal parameters (mass median diameter (MMD) and count median diameter (CMD)) were


determined for each of the BC size distributions by representing them using mono-modal log normal fit (e.g., Shiraiwa et al., 2007; Schwarz et al., 2008; Liu et al., 2010; Wang et al., 2016) of the following form:

$$\frac{dA}{d\ln D_p} = \sum \frac{A_0}{\sqrt{2\pi}\ln\sigma_m} \exp\left[-\frac{\left(\ln D_p - \ln D_m\right)^2}{2\ln\sigma_m}\right]$$

(1)

Here $A_0$ corresponds to mass/number concentration of the mode, $D_m$ is the mass/count median diameter, $D_p$ is particle diameter and $\sigma_m$ is the geometric standard deviation. A typical mass and number size distribution and corresponding modal fit are shown in Figure 4(a), and the temporal variations of daily mean MMD (CMD) values derived from individual size distributions are shown in Figure 4(b) and Figure 4(c).  Here each symbol represents the mean value for the day and the

vertical line passing through it is the corresponding standard deviation. The solid continuous line shows the 30-day smoothed variation and the dotted vertical lines separate different seasons. Both, MMD and CMD, gradually increase from their respective minimum values during the summer monsoon season (SMS) through post monsoon and winter season to reach the highest values during the pre-monsoon season, indicating the progressively dominating share of larger particles in the size distribution. Further, the seasonal mean mass and number size distributions are shown in Figure 5 highlighting the changes in

the BC microphysical properties (abundance and sizes) with changes in the source processes along with seasonal transformation.  These size distributions were parameterized by evolving least-squares fitting to mono- modal log-normal distribution. These modal parameters estimated for different seasons were tabulated in Table-2.

Previous studies suggested that different sources emit BC particles with different core diameters, where smaller modes indicate urban outflow with dominance of fossil fuel sources, and larger modes (>0.20 µm) are more likely to be associated

with solid fuel sources including biomass/coal burning (Schwarz et al., 2008; Liu et al, 2010, 2014; Sahu et al., 2012; Reddington et al., 2013).

Viewed in the light of the above, the lowest seasonal mean MMD ~ 0.169 ± 0.013 µm (CMD ~ 0.090 ± 0.005 µm), occurring during SMS, highlights the possible dominance of fresh emissions containing smaller sized particles (and/or externally mixed particles), when washout also is quite significant. As the season advances, MMD and CMD increase, due to aging processes

(including coagulation of the agglomerates), as the removal mechanism is weakened (significantly low precipitation) during these seasons. The modal values during the post monsoon and winter seasons (MMD ~ 0.182 ± 0.012; ~ 0.193 ± 0.017 µm and CMD ~0.100 ± 0.006; ~ 0.111 ± 0.006 µm respectively) are comparable with those reported for reported from the continental outflows (McMeeking et al., 2010, 2011; Ueda et al., 2016) suggesting mixed sources and/or aged BC. By the pre-monsoon season, the MMD values reached beyond 0.20 µm and continued to remain so throughout PMS highlighting the

dominance of larger core BC particles (likely solid fuel, e.g., coal/biomass burning emissions). These seasonally changing size distributions (and MMD values) reflect the combined effect of the nature of sources, efficiency of sink and role of transport. It is difficult to delineate local sources from those in far field regions that transport BC to the receptor sites.  While





the smaller MMD during the SMS suggest the effective wet removal larger sized BC particles, while weaker wet removal in the other seasons led to larger MMD values. As evident from the supplementary figure S1, intense fire events that occur over the Indian region, and this combined with enhanced boundary layer heights (due to increased temperatures) enabling effective dispersion of pollutants, both horizontally and vertically, is responsible for larger BC cores observed during the

PMS.

Table-2 and Figure 5(a) also suggested similar conclusions. The modal diameters changed from lowest values (~ 0.165 µm) during SMS to the highest value (~ 0.214 µm) in the PMS through the seasons PoMS (~0.178 µm) and winter (~0.191 µm), signifying the changing source and atmospheric dynamical processes and that size distributions progressively developed towards larger sized BC particle dominance from SMS to PMS. The peak concentrations suggested the highest mass loading

in winter ($M_0$ ~ 4679 ng m$^{-3}$), followed by PoMS ($M_0$ ~ 2752 ng m$^{-3}$) and SMS ($M_0$ ~ 2225 ng m$^{-3}$), whereas the lowest loading is seen PMS ($M_0$ ~ 2031 ng m$^{-3}$).

The size distribution during SMS is slightly skewed towards lower size regime and resulted in a broad width (standard deviation ($\sigma_m$) ~1.99), and as the seasons progress towards PMS such skewness (or lack of it as indicated by decreased $\sigma_m$ values) progressively shifted towards larger size ranges in addition to drop in smaller sized particle concentrations due to the

changes in the nature of sources. These values also indicate the other atmospheric dynamical processes that control the BC life cycle apart from the source and sink as described above. Extended rainfall reduces the overall mass loading due to wet scavenging in SMS and partly in PoMS, smaller BC particles prevailed while larger particles are more prone to being scavenged. Lack of wet removal mechanism in dry seasons (winter and PMS) resulted in extended life time of BC resulting in greater atmospheric life time. Also, seasonally varying boundary layer dynamics determines the extent of the near surface

aerosol loading (this aspect is discussed in a later section).

Similar to the mass size distributions, examination of the modes of the number size distribution in different seasons suggested smaller size particles (mode ~0.080 µm) during SMS which progressively increased towards larger core BC particles with season from PMS (~0.130 µm) through PoMS (~0.099 µm) and winter (~0.112 µm) with broader widths, highlighting seasonal changes in the nature of dominant sources. But mean BC number size distributions showed a slightly

different picture when mode peak values are considered. The largest modal number concentration is seen during SMS ($N_0$ ~ 1502 cm$^{-3}$) (albeit lower mode diameter suggesting the smaller particle dominance possibly of regional origin due to the extended precipitation causing below-cloud scavenging restricting the abundance and life time of larger particles), followed by winter ($N_0$ ~ 1270 cm$^{-3}$) and PoMS ($N_0$ ~ 1219 cm$^{-3}$). PMS showed the lowest peak number abundance ($N_0$ ~ 407 cm$^{-3}$).

A summary of MMD values reported in the literature with different emission sources and atmospheric conditions along with

the present values is made in Table-3. Several earlier publications reported a range of MMD values (0.100-0.170 µm) for urban regions with near source fossil fuel emissions (McMeeking et al., 2010; Liu et al., 2014; Laborde et al., 2013; Cappa et al., 2012; Kondo et al., 2011; Cheng et al., 2018), whereas, urban/continental outflows depict MMDs in the range of 0.140–0.180 µm (Shiraiwa et al., 2007; Wang et al., 2018). The MMD values are ~0.211 ± 0.014 µm for fresh biofuel/crop residue sources (Raatikainen et al., 2017), and in the range ~0.220-0.240 µm for aged BC from biomass burning sources (Liu et al,.



2010) or high urban pollution episodes with high biomass burning (Gong et al., 2016). Table-3 highlights that MMD values depend strongly on the source processes and reiterate that present values depict seasonally changing sources, from fossil fuel dominance in SMS to solid fuel (biomass or coal burning) dominance in PMS through mixed sources typical for outflow during PoMS and winter.

**3.3 Seasonal changes in mixing state of BC**

The mixing state of BC depends on several parameters such as concentration of condensable species that adsorb or condense on to the BC core, atmospheric humidity, the atmospheric lifetime of BC cores (including photochemical aging) and the size distributions (Liu et al., 2013; Ueda et al., 2016; Cheng et al., 2018). As the source strengths of condensable species are likely to vary with season (due to seasonality of local emissions, prevailing meteorology and long-range transport) the mixing state of BC would respond to such changes. With a view to examining this, we have quantified the mixing state of BC in terms of (a) the relative coating thickness (RCT ~ $D_p/D_c$) and (b) absolute coating thickness (ACT ~ $(D_p-D_c)/2$), where $D_p$ and $D_c$ respectively represent the particle diameter and the core diameter. Both these parameters have been determined from BC mass equivalent diameters which depend only on the emission source characteristics and not on the morphology or mixing state of the particles. They help to evaluate the changes in physiochemical properties of BC during its atmospheric transit life time. While RCT quantifies the extent of coating on a BC core, ACT provides the magnitude of coating in nm. The temporal variation of daily mean values of RCT (half circles) and ACT (star symbol) are shown in Figure 6, where the solid line represents the 30-day running-mean smoothed variation revealing the seasonality.

The figure shows strong seasonality in coating on BC owing to the effects of the multiple processes discussed above. Both relative and absolute coating thicknesses are very low during the monsoon season, and increase gradually towards winter through the post monsoon (though the increase is not fully depicted due to a gap in the data). Further the values slightly dropped during February-March (seasonal transformation from winter to PMS) and further increased again towards summer. The low values of RCT and ACT during SMS (seasonal mean RCT ~1.16 ± 0.04 and ACT ~12.12 ± 4.98 nm) indicate thin coating on freshly emitted small BC cores. This is attributed to the short life time of BC in this season due to their efficient washout by the wide-spread monsoon rainfall and the lower concentration of condensable gas phase precursors caused by this wet removal, so that the cores are basically of nascent BC particles, which emanate from fossil fuel emissions, and have the least mass/ number median diameters (Figure 5, black line). As the season advances to PoMS, the monsoon activity is subdued leading to increased lifetime of BC, and a change in airmass (Figure 2b) results in advection of different types of aerosols and gaseous species from the more polluted north-western IGP (rather than the predominantly oceanic nature of the airmass in SMS). The airmass is drier than during monsoon. The particle size distribution shows increased presence of larger particles, and consequently higher median diameters (Figure 5, red lines) indicating change in nature of sources. All these resulted in an increase in overall coating during PoMS with seasonal mean values of 1.32 ± 0.14 for RCT and 28.74 ± 12.31 nm for ACT highlighting an enhancement of ~ 32 % of the particle sizes due to thick coating of condensable vapours on BC



cores during this season. Intra-seasonal variability (highlighted by the standard deviation of the seasonal mean values of RCT and ACT in Figure 6) is also higher during PoMS. During winter, in general the RCT values are higher (mean RCT ~1.34 ± 0.12) the extent of absolute coating on the cores is also highest (mean ACT ~ 33.51 ± 11.76 nm) suggesting thickly coated BC particles. This is attributed to the availability of condensable vapours advected by the continental airmasses (Figure 2c)

and the longer residence time of BC and the larger mode diameters (Figure 5, blue line). In winter, the airmass pathways originated from the highly polluted IGP region prevail over Bhubaneswar with abundance of condensable species. The combination of prevailing calm weather conditions and absence of precipitation enhanced the life of aerosols and thus resulted in thickly coated BC aerosols. As the season changes to summer/ pre-monsoon, the RCT decreased (mean RCT~1.26 ± 0.10) because of the larger BC cores (highest MMD values, green line in Figure 5) while the absolute coating

thickness remains high (mean ~27.41 ± 10.72 nm). This occurs as a result of the relative increase in larger particles, which for the same RCT would lead to higher ACT.

The seasonality of the coating characteristics, discussed above, is further demonstrated in Figure 7, which shows the frequency distribution of the RCT and ACT values for each season (median values for each season are also shown in the figure). The monsoon season is characterised by a narrow distribution, with the lowest median values for both RCT and

ACT. Such thin coatings are due to the reduced life-time and absence of multiple processes which include: (a) reduced life-time, leading to not having enough time to get thickly coated (as coating thickness depends on the particle life time also), (b) reduced concentration of advected anthropogenic precursor species, making lesser availability of coating material, and (c) fast wet-removal (of the core and the coating material) by extensive rainfall. South-westerly/westerly air masses prevailed during this period that advect cleaner marine air to the site, while wash-out due to large-scale precipitation removed aged

aerosols. During other seasons, the distributions are quite broad, and showed multiple maxima (especially during PoMS and winter). During these seasons, BC particles in the IGP outflow are characterised by extensive coating resulting from the large shift in the airmass from marine to continental at the end of the monsoon season bringing air from the largely polluted central IGP, added with the associated change in the size distribution and residence time. Based on their study during a heavy air pollution episode in Shanghai, Gong et al., (2016) have reported instantaneous ACT values ranging between 50-300 nm in

different number size regimes with distinct sources and aging process of BC. Such high magnitudes of ACT are possible in the extremely polluted airmasses in the near the immediate vicinity of sources. Coating on BC particles enlarges the available absorption cross-section and results in absorption enhancement. Moffet and Prather (2009) have examined the sensitivity of optical properties to the microphysical properties of BC, and found absorption enhancement in the ranges from 1 (for no coating) to 3.4 (for largest particle sizes), and further, larger absorption enhancements (1.6) for larger shell/core ratios (RCT

~1.75) obtained for the aged BC, which compared to those for fresh BC (absorption enhancement of 1.4 and RCT ~1.07). Direct comparison of the coating parameters (RCT and ACT) in the present study with other studies is not possible. This is because of the difficulties in comparison across studies as detailed by Cheng et al., (2018), which include different system configurations, difference in techniques used in fitting of scattering amplitudes and the range of mass equivalent diameters.





### 3.4 Diurnal variations of rBC mass concentration and RCT

At the shorter times scales (within a day), the mesoscale processes and atmospheric boundary layer (ABL) dynamics would be influential in modulating the mixing characteristics of BC (for example, Liu et al., 2014; Laborde et al., 2013). To examine this, the seasonal mean diurnal variations of BC mass concentration (red-star symbol) and relative coating thickness (RCT) (blue-filled circle) are shown in Figure 8. The vertical lines in each panel mark the local sunrise and sunset times for the season.

The figure reveals, in addition to the typical double-humped diurnal variation of BC mass concentration due to the well-known combined effects of boundary layer dynamics (Kompalli et al., 2014) and diurnal variation of the anthropogenic activities, very interesting links between BC core and relative coating thickness. While BC and RCT depict the double-humped diurnal variation, they are almost in the opposite sense, and the amplitude of the variation has a marked seasonality, with highest amplitude in winter (when the day-night temperature variation is the highest; with $\Delta T$ (i.e. $T_{max}$-$T_{min}$) ~ 12 °C where $T_{max}$ and $T_{min}$ are maximum and minimum temperatures over the 24 hour period) and lowest amplitude during the monsoon (when the temperature variation is the least; $\Delta T$ ~ 4.9 °C). The diurnal variation in BC mass concentrations and the factors determining it over the Indian region have been widely reported (e.g., Beegum et al., 2009; Mahapatra et al., 2013a; Kompalli et al., 2014), the diurnal pattern of the BC mixing state has not been previously observed. More intriguing is the sense of variation opposite to that of BC; with peaks occurring around 02:00 to 04:00 hrs and 12:00 to 15:00 hrs local time with troughs in between.

As previously discussed, the increased RCT values are associated with aging of BC cores and availability of condensable vapours (which are generally co-emitted with BC or produced photochemically from species that are co-emitted). As a result, the peak in RCT during daytime can be attributed to the abundance of condensable material originating due to photochemistry and thus gas-phase photochemical processing leading to enhancement in the extent of coating (Liu et al., 2014; Chakraborty et al., 2018; Brooks et al., 2018). The second peak occurring during late night- early morning periods is more likely to be linked to the increased aging of BC, (lack of fresh emissions on the one hand and reduced condensation sink due to decrease in concentration of pre-existing foreign particles that compete with BC to adsorb condensable vapours (e.g., Babu et al., 2016 and references therein) on the other). The amplitude of the day time peak in RCT is greater than or equal to the early morning (dawn) peak due to two factors: (a) enhanced dispersion during day time due to increased convective mixing results in reduced particle abundance, thus increasing the probability of enhanced vapour adsorption on individual particles, (b) day time build-up of possible coating material due to photochemistry (which is a stronger factor than the first one), and both these conducive conditions are not available during early morning. ACT also showed a similar pattern to RCT but much a more pronounced diurnal variation, whereas there is no discernible variation in the diurnal pattern of MMD (supplementary Figure S2).

The morning peak in BC mass concentration, occurring shortly after sunrise is due to the well-known 'fumigation effect', i.e., when the thermals generated after the sunrise break the inversion and bring down the pollutants from the residual





layer, as has been discussed in several papers (Beegum et al., 2009; Kompalli et al., 2014; Babu et al., 2016). Also rush hour concentration due to build-up of vehicular traffic also contributes to this. The succeeding trough is due to enhanced convective mixing and deepening of the ABL. After the sunset as the thermals subside, the shallow nocturnal boundary layer sets in and the resulting stable conditions lead to second peak due to confinement of the aerosols near to the surface. Further,

lower temperatures and wind speeds coupled with reduced emissions result in gradual decrease in BC mass concentration leading to a night time minimum. Interestingly when the morning peak in BC mass concentration is observed due to the combined effect of the boundary layer dynamics (fumigation effect) and sources (rush hour traffic contribution), RCT showed a minimum suggesting that the rush hour traffic source which would push up the BC concentration and lower the RCT outweighs the fumigation effect though both may be occurring. The diurnal variation is more pronounced during the

PoMS and winter (Fig. 8b,c) and subdued during the SMS and the PMS (Fig 8a and 8d) owing to varying strength of ventilation of aerosols due to changes in the atmospheric boundary layer dynamics in different seasons (Kompalli et al., 2014). Similarly, the amplitude in the diurnal variation of RCT is highest in the PoMS (RCT changing from 1.42 to 1.25), followed by winter and largely subdued during the SMS and PMS. Not only are diurnal variations in RCT suppressed in the SMS and PMS compared to the winter and PoMS due to differences in boundary layer dynamics, in SMS less aged BC and

enhanced precipitation removing coating substances and in the PMS the prevailing larger BC core sizes reducing the relative coating thickness may also play a role.

**3.5 Non-refractive PM1.0 mass concentrations**

To examine the likely coating material on BC cores, mass concentration and chemical composition of non-refractive PM1.0 (NR-PM1) aerosols obtained from the measurements of the ACSM are examined.

The seasonal mean mass concentrations and fractional contribution of different species (organics, sulphate, nitrate, ammonium, and chloride), as deduced from the ACSM measurements are shown in Figure 9. There are clear seasonal changes in chemical composition associated with distinct airmasses and wide variety of sources. While as expected, the mass concentration was highest during winter ($20.45 \pm 22.55$ µg m$^{-3}$) followed by PoMS ($13.90 \pm 10.62$ µg m$^{-3}$) (due to the combined effects of reduced removal, confinement of aerosols due to shallow boundary layer and change in long-range

transport), the mass fraction (MF) showed a dominant contribution of organics (0.39-0.49), with the highest fraction in winter (0.49). Sulphate, the next major contributor (MF varied in the range 0.27 to 0.47) showed strong seasonality, being highest in the PMS (0.47) followed by the SMS (0.41), and lowest in winter (0.27) and PoMS (0.28). This is further corroborated by the airborne Aerosol Mass Spectrometer (AMS) measurements during SWAAMI (Brooks et al., 2018), which have shown significant presence of sulphate in the Central IGP extending to higher altitudes even during the monsoon

season. Ammonium, nitrate and chloride are only minor components of the NR-PM1 mass loading. The significant presence of nitrate during the PoMS and winter (14%) suggest advection of anthropogenic emissions from the central IGP likely as a result of enhanced ammonia emissions during the growing season and cooler temperatures, favouring $NH_4NO_3$ formation.





It is clear that when the IGP air masses prevailed (PoMS and winter) organics dominated the NR-PM1 mass concentration, while during mixed/coastal air masses (SMS and PMS) sulphate also equally or prominently contributed to NR-PM1 mass concentration, clearly depicting seasonal contrast in the mass concentrations with the changing nature of sources in distinct airmasses.

Earlier studies (e.g. Kumar et al., 2016; Thamban et al., 2017; Chakraborty et al., 2018 and references therein) have examined the NR-PM1 chemical composition over Kanpur, an urban location in the central IGP, using an aerosol mass spectrometer (AMS) and reported the dominance of organics during PoMS and winter. Pandey et al. (2014) developed a multi-pollutant emission inventory for different sectors of India and reported that residential biomass burning (cooking stoves) is the largest contributor for PM2.5 and organic carbon aerosols. Recently from the molecular analysis of the PM2.5

emissions over a village in the IGP, Fleming et al., (2018) have characterized a wide range of particle phase compounds produced by traditional cook stoves and pointed out various organic compounds originate from these sources. Viewed in this context, dominance of organics in the IGP outflow is not surprising.  From their filter based chemical composition measurements over Bhubaneswar, Mahapatra et al., (2013b) have suggested that the sources of $SO_4^{2-}$ were anthropogenic, crustal and marine, with the major contributor being the anthropogenic sources. So sulphate is possibly of mixed origin and

present in significant proportions, more so during non-IGP airmass periods (SMS and PMS).

### 3.6  Diurnal variations of NR-PM1 species and association with BC mixing state

It is worthwhile to examine the diurnal pattern of NR-PM1 species which will act as coating substances and understand any possible association with that of RCT.  In order to evaluate this in terms of the relative magnitude of each species, hourly averaged mass fractions of organics, sulphate, nitrate, ammonium and chloride aerosols are considered and

the seasonal mean diurnal variation of these are shown in Figure 10. It is clearly seen that, sulphate dominated during the daytime in the PoMS, winter and the PMS, with a diurnal variation that resembled that of the RCT (Figure 8) during these seasons. The diurnal variation indicates strong photo-chemical production of sulphate from gas-phase chemistry. The weak nature of the day time peak in sulphate during the PMS may be attributed to enhanced dispersion resulting in lower near surface concentrations which overcomes photochemical production. The organics dominated during the night, which is due

to a combination of factors including source processes and photochemistry, and its diurnal variation is almost opposite to that of the RCT.  The diurnal variations of organics depict two pronounced peaks occurring during morning (06:00-08:00 hrs) and late evening (20:00-22:00 hrs), similar to the rBC mass loading.

The diurnal pattern of other species (nitrate, ammonium and chloride, whose concentrations were lower except in winter) followed a pattern similar to organics, but with less variation. Absence of day time enhancement of nitrate and

ammonium indicated that photochemical production may not be significant or possibly destruction (evaporative loss, i.e., gas-particle partitioning of $NH_4NO_3$) dominated.





Even though it is very difficult to determine the coating material on the BC particles that exist in a multi-component organic, inorganic aerosols, gaseous vapours system, the above discussion suggests two contrasting possibilities of coating material on BC:  (a) concurrent peaks in RCT (as seen in Figure 8) and sulphate during the diurnal cycles indicates that the most probable material for the coating is sulphate,  whereas  (b) possibility of organic matter acting as coating material cannot be ruled out because observed minima in the mass fraction of organics during RCT peaks (during dawn and afternoon hours) suggested a possible loss of organic vapours through condensation on large number of pre-existing BC particles, thereby contributing to their coating. Boundary layer dynamics and source processes play an important role not only on particle loading but also in determining the coating (Liu et al., 2014; Gong et al., 2016; Thamban et al., 2017; Wang et al., 2018). Increased ventilation during day time due to enhanced boundary layer heights dilutes aerosol concentrations, thereby reduces competition among particles for adsorption of condensable vapours. Further, freshly produced particles (with less or no coating) from primary as well as secondary sources are greater during day in general and this enables more efficient adsorption condensable species on these particles, compared to relatively aged particles during night which already more internally mixed. Further, the distinct nature of sources of various species is also an important factor.  Majority of the BC and organic aerosol are produced in locations away from the sulphate sources.  At night in a collapsed ABL with stable conditions the sampling station sees local sources which are predominately consists of BC and organics with reduced sulphate, whereas during the day in a well-developed ABL, both near field and far field source contributions mix together. This results in dilution of the contribution of species from near field sources to coating, but will introduce increased contribution from species from far field sources. This changes the balance of sulphate to organics mass concentrations and also the RCT of the BC.

Our observations indicate enhanced sulphate occurs due to photochemistry. In addition, the possibility of organic matter acting as a coating material is not ruled out since secondary organic aerosol is known to have photochemical origin (Chakraborthy et al., 2018). Thamban et al., (2017) have reported an increase in oxygenated organic species during the day time with a diurnal trend similar to the fraction of thickly coated BC.

**3.7  Seasonality of the association between rBC relative coating thickness and NR-PM1 chemical species**

Further, we examined the association between the mass fractions (MF) of different species with simultaneous RCT values by considering hourly mean values of both the parameters. The association between hourly mean RCT and MF of organics and sulphate (the dominant NR-PM1 species) is shown in Figure 11 (other species did not show any perceptible association). The colour bar indicates the percentage of occurrence of a particular value of RCT for a corresponding MF value of the species considering the entire data set for that season.  During SMS (Figure 11a &Figure 11e), since there are very few available simultaneous observations of RCT and MF no conclusion about their association can be drawn, and also the extent of coating is very much reduced during this season.  During PoMS (when the IGP outflow airmasses prevailed), as seen from the Figure 11b & Figure 11f, instances of higher RCT decreased with increasing MF of organics, whereas the association is vice versa



between RCT and MF of sulphates. This suggests that sulphates may be the possible preferential coating substance, as increasing fractions of sulphates in the total mass concentrations contributed to the enhanced coating on BC particles.

During winter (Figure 11c & Figure11g) similar to the PoMS increasing MF of organics has a negative correlation with RCT, whereas the MF of sulphate did not show any clear association. As the season changes to PMS, the association between RCT and MF is reversed to what it was during PoMS, with the population of highly coated particles decreasing with increasing MF of sulphate, while RCT increased with increasing MF of organics. It is known that the nature of initial coating and mixing state of BC particles is dependent on the type of BC sources (Liu et al., 2013) and also on the nature of prevalent semi-volatile vapours and heterogeneous interactions with gas-phase species that act as condensable material. The observed association of organics and sulphate with RCT suggests possible preferential coating, which is not dependent on the mass loading of the dominant species in the PM1.0, but rather dependent on the nature of dominant sources (gaseous precursors from the similar sources that produce BC are important). Extent of coating depends more on the strength of the sources, number/surface area size distribution of the particles and concentration of condensable vapours coupled with atmospheric dynamical processes.

As discussed in the previous sections, BC in the highly polluted IGP outflow is characterized by higher mass loadings and mixed sources (MMD ~0.180-0.190 µm) which include vehicular, industrial emissions (fossil fuel sources) and wide spread thermal power plants over the IGP (Thamban et al., 2017; Brooks et al., 2018) that co-emit gaseous $SO_2$ along with BC. Enhanced RCT with increased MF of sulphates indicate the possibility that sulphate resulting from the vapour phase chemistry of $SO_2$ emissions may be an important condensable species on BC particles during their extended atmospheric transit in the outflow (Takami et al., 2013; Miyakawa et al., 2017). Larger BC cores (MMD ~ 0.200-0.220 µm) during the pre-monsoon indicate that solid fuel sources (including biomass/coal burning processes) which also emit organic material (vapours as well as particulates) along with BC, and sulphate in primary particulate form (Pandey et al., 2014; Fleming et al., 2018). As seen from the supplementary figure S1, increased fire counts during the PMS indicate sources of significant amounts of organic material apart from BC. This, combined with enhanced dilution of the species due to ABL dynamics modulating both particle and condensable species concentrations during the atmospheric transit contribute to positive association between RCT and MF of organics. Such positive association suggest that organic vapours possibly contributed to the enhanced coating on BC during the PMS.

While the present work highlighted the microphysical properties of the refractive BC aerosols and brought out the difference between the IGP outflow and other airmass regimes, further investigations (both experimental and theoretical) are needed to ascertain the possible radiative (including absorption enhancement) and climatic implications due to the observed microphysical properties, extent of coating and changes in the mixing state of the BC due to various host coating materials. This will form the focus of future work.

## 4    Summary and Conclusions



The present study has determined the mass concentration, size distributions and mixing state of refractive BC particles from the single particle soot photometer observations carried out over Bhubaneswar located in the eastern coast of India. Our important findings are as follows.

(1) The rBC mass concentration are higher during winter (~1935 ± 1578 ng m$^{-3}$), followed by post-monsoon (~1338 ± 1396 ng m$^{-3}$). Reduced rainfall, calm wind conditions, coupled with decreased ventilation due to the shallow boundary layer resulted in such build-up of aerosols. Lowest rBC mass loading (~816 ± 835 ng m$^{-3}$) is seen during the pre-monsoon, possibly due to enhanced convective mixing leading to significant dispersion of the near surface aerosols.

(2) BC size distributions indicated seasonally changing nature of sources with smaller BC cores (MMD ~0.150-0.170 µm) in the summer monsoon highlighting fossil fuel sources to larger BC cores (MMD > 0.210 µm) in the pre-monsoon suggesting the prominence of solid fuel sources. rBC that originated from mixed sources (both fossil fuel and solid fuel) (MMD ~0.190-0.195 µm) prevailed when the airmass pathways originated from the highly polluted IGP region.

(3) Further, the IGP outflow is characterized by the highly coated BC particles with bulk relative coating thickness (RCT) in the range ~1.3-1.8 and absolute coatings of 50-70 nm on the BC cores. Abundance of condensable species, combined with prevailing calm weather conditions and absence of precipitation resulted in extended life time, and thus thickly coated BC particles. During the SMS efficient wet scavenging restricts the life time of aerosols and results in the lowest coatings observed throughout the year (median ACT ~12.35 nm and RCT ~1.15), indicating relatively nascent BC aerosols. During the PMS, significantly coated (RCT ~1.2-1.3) and larger core BC particles prevailed which may have important regional climatic implications.

(4) BC particles with relatively thicker coating are noticed during day time in all the seasons, which is due to the abundance of photo-chemically produced condensable species, and thus gas-phase photochemical processing. The diurnal amplitude is highest in winter and lowest in the SMS and this highlighted the role played by ABL dynamics in modulating rBC microphysical properties.

(5) Diurnal variation of sulphate resembled that of the RCT of rBC with a clear day time dominance in the PoMS, winter and SMS, indicating strong photo-chemical production of sulphate from gas-phase chemistry. During the PMS, the day time peak in sulphate is weak which may be attributed to enhanced dispersion resulting in lower near surface concentrations which overcomes photochemical production. Diurnal variation of the organics resembled that of BC mass concentrations with typical double maxima.

(6) Examination of diurnal variations presented two contrasting possibilities of coating material on BC: (a) sulphate acting as the most probable material coating the BC core due to its abundance during the day time, and (b) organics acting as condensable species, as indicated by the observed minima in mass fraction of organics during RCT peaks suggesting a possibility of loss of organic vapours through condensation on large number of pre-existing BC particles, thereby contributing to their coating.



(7) Examination of NR-PM1 mass fractions in conjunction with BC coating thickness suggest that the coating on BC is positively associated with sulphates during the IGP outflow (March to September) while the association is more stronger with organics during PMS when coastal airmasses prevailed; thereby highlighting preferential coating in different seasons with conducive species availability through advection .

Our study provides insight into the seasonally varying source processes and changes in the microphysical properties of BC over Bhubaneswar and highlights the delineation between the IGP outflow and the non-IGP airmasses. Further investigations are needed to understand the sensitivity of the optical and hygroscopic properties of BC to such seasonally varying microphysical properties and atmospheric processing of BC over the Indian region.

**Data availability**

Data are available upon request from the contact author, S. Suresh Babu (s_sureshbabu@vssc.gov.in).

**Competing interests**

The authors declare that they have no conflict of interest.

**Author contributions**

SSB, SKS, KKM and HC conceptualized the experiment and finalized the methodology. SKK, TD and RB are responsible
for the maintenance and operation of the SP2 and the ACSM. SKK carried out the scientific analysis of the data supported by MF, DL, ED, JB and JA.  SKK drafted the manuscript.  SSB, KKM, SKS and HC carried out the review and editing of the manuscript.

**Acknowledgements**
This study was carried out as part of collaborative "South West Asian Aerosol Monsoon Interactions (SWAAMI)" experiment under a joint Indo-UK (NERC) project namely "Drivers of variability in the South Asian Monsoon" under the "National Monsoon Mission (NMM)" of the Ministry of Earth Sciences (MoES), Government of India, in which the ISRO, the Indian Institute of Science (IISc), Bengaluru and the University of Manchester, UK are partners. Bhubaneswar station is a supersite set-up under SWAAMI, for long term characterization of the IGP outflow. It also is a part of the network under the
Aerosol Radiative Forcing over India (ARFI) project of the Indian Space Research Organisation-Geosphere Biosphere Program. The authors are thankful to the Director, Institute of Minerals and Materials Technology (CSIR-IMMT) for the support. We acknowledge NOAA Air Resources Laboratory for the provision of the HYSPLIT transport and dispersion model and READY website (http://www.arl.noaa.gov/ready.html) used in this study. We acknowledge the use of data and





imagery from LANCE FIRMS operated by NASA's Earth Science Data and Information System (ESDIS) with funding provided by NASA Headquarters (http://earthdata.nasa.gov/firms).

**5**      **References**

Adachi, K., Y. Zaizen, M. Kajino, and Y. Igarashi: Mixing state of regionally transported soot particles and the coating effect on their size and shape at a mountain site in Japan, J. Geophys. Res. Atmos., 119, 2014, 5386–5396, doi:10.1002/2013JD020880.

Adachi, K., Chung, S. H., and Buseck, P. R.: Shapes of soot aerosol particles and implications for their effects on climate, J.
**10**      Geophys. Res.-Atmos., 115, D15206, doi:10.1029/2009JD012868, 2010

Asnani, G.C., Tropical Meteorology, Vol.1 and Vol.2, 1012 pp, 1993. Indian Institute of Tropical Meteorology, Pashan, Pune.

Babu, S.S., Kompalli, S.K. Moorthy, K.K.: Aerosol number size distributions over a coastal semi urban location: Seasonal changes and ultrafine particle bursts. Sci.Total Environ., 563-564, pp 351–365,2016.
**15**      http://dx.doi.org/10.1016/j.scitotenv.2016.03.246.

Babu, S.S., Manoj, M.R., Moorthy, K.K., Gogoi, M.M.,Nair V.S., Kompalli, S.K., Satheesh, S.K., Niranjan, K., Ramagopal, K., Bhuyan, P.K. and Singh, D.: Trends in aerosol optical depth over Indian region: Potential causes and impact indicators. J. Geophys. Res., 118, 11: 794-11,806, 2013.

Beegum, S.N., Moorthy, K.K., Babu, S.S., Satheesh, S.K., Vinoj, V., Badarinath, K.V.S., Safai, P.D., Devara, P.C.S., Singh,
**20**      S.N., Vinod, Dumka, U.C., Pant, P.: Spatial distribution of aerosol black carbon over India during premonsoon season. Atmos. Environ., 43: 1071–1078, 2009.

Bond, T. C., Doherty, S. J., Fahey, D. W., Forster, P. M., Berntsen, T., DeAngelo, B. J., Flanner, M. G., Ghan, S., Karcher, B., Koch, D., Kinne, S., Kondo, Y., Quinn, P. K., Sarofim, M. C., Schultz, M. G., Schulz, M., Venkataraman, C., Zhang, H., Zhang, S., Bellouin, N., Guttikunda, S. K., Hopke, P. K., Jacobson, M. Z., Kaiser, J. W., Klimont, Z.,
**25**      Lohmann, U., Schwarz, J. P., Shindell, D., Storelvmo, T., Warren, S. G., and Zender, C. S.: Bounding the role of black carbon in the climate system: A scientific assessment, J. Geophys. Res. Atmos., 118, 5380–5552, doi:10.1002/jgrd.50171, 2013.

Bond, T. C. and Bergstrom, R. W.: Light absorption by carbonaceous particles: An investigative review, Aerosol Sci. Tech., 40, 27–67, doi:10.1080/02786820500421521, 2006.
**30**  Brooks, J., Allan, J. D., Williams, P. I., Liu, D., Fox, C., Haywood, J., Langridge, J. M., Highwood, E. J., Kompalli, S. K., O'Sullivan, D., Babu, S. S., Satheesh, S. K., Turner, A. G., and Coe, H.: Vertical and horizontal distribution of sub-





micron aerosol chemical composition and physical characteristics across Northern India, during the pre-monsoon and monsoon seasons, Atmos. Chem. Phys. Discuss., https://doi.org/10.5194/acp-2018-1109, in review, 2018.

Cappa, C. D., Onasch, T. B., Massoli, P., Worsnop, D. R., Bates, T. S., Cross, E. S., Davidovits, P., Hakala, J., Hayden, K. L., Jobson, B. T., Kolesar, K. R., Lack, D. A., Lerner, B. M., Li, S.-M., Mellon, D., Nuaaman, I., Olfert, J. S., Petäjä, T., Quinn, P. K., Song, C., Subramanian, R., Williams, E. J., and Zaveri, R. A.: Radiative Absorption Enhancements Due to the Mixing State of Atmospheric Black Carbon, Science, 337, 1078–1081, doi:10.1126/science.1223447, 2012.

Chakraborty, A., Mandariya, A.K., Chakraborti, R.,Gupta,T., Tripathi, S.N.: Realtime chemical characterization of post monsoon organic aerosols in a polluted urban city: sources, composition, and comparison with other seasons, Environ. Pollu., 232, 310-321, 2018. https://doi.org/10.1016/ j.envpol.2017.09.079.

Cheng,Y., Li,S.M., Gordon,M.,and Liu,P.:Size distribution and coating thickness of black carbon from the Canadian oil sands operations.Atmos. Chem. Phys., 18, 2653–2667, 2018.

China, S., Mazzoleni, C., Gorkowski, K., Aiken, A. C., and Dubey, M. K.: Morphology and mixing state of individual freshly emitted wildfire carbonaceous particles, Nat. Commun., 4, 2122, doi:10.1038/ncomms3122, 2013.

Dey, S., S. N. Tripathi, and S. K. Mishra: Probable mixing state of aerosols in the Indo-Gangetic Basin, northern India, Geophys. Res. Lett., 35, L03808, 2008.

Fleming, L.T., Lin, P., Laskin, A., Laskin, J., Weltman, R., Edwards, R.D., Arora, N.K., Yadav, A., Meinardi, S., Blake, D.R. & Pillarisetti, A.:. Molecular composition of particulate matter emissions from dung and brushwood burning household cookstoves in Haryana, India. Atmos. Chem. Phys., 18(4), pp.2461-2480, 2018.

Gautam, R., Hsu, N.C., Lau, K.M., Tsay, S.C., and Kafatos, M., Enhanced premonsoon warming over the Himalayan-Gangetic region from 1979 to 2007.Geophys. Res. Lett., 36, L07704, 2009.

Gong, X. D., Zhang, C., Chen, H., Nizkorodov, S. A., Chen, J. M., and Yang, X.: Size distribution and mixing state of black carbon particles during a heavy air pollution episode in Shanghai, Atmos. Chem. Phys., 16, 5399–5411, 2016.

Huang, X. F., Gao, R. S., Schwarz, J. P., He, L. Y., Fahey, D. W., Watts, L. A., McComiskey, A., Cooper, O. R., Sun, T. L., Zeng, L. W., Hu, M., and Zhang, Y. H.: Black carbon measurements in the Pearl River Delta region of China, J. Geophys. Res.-Atmos., 116, D12208, doi:10.1029/2010jd014933, 2011.

Huang, X. F., Sun, T. L., Zeng, L. W., Yu, G. H., and Luan, S. J.: Black carbon aerosol characterization in a coastal city in South China using a single particle soot photometer, Atmos. Environ., 51, 21–28, doi:10.1016/j.atmosenv.2012.01.056, 2012.

IPCC, 2013: Climate Change., The Physical Science Basis. Contribution of Working Group I to the Fifth Assessment Report of the Intergovernmental Panel on Climate Change (Stocker, T.F., D. Qin, G.-K. Plattner, M. Tignor, S.K. Allen, J. Boschung, A. Nauels, Y. Xia, V. Bex and P.M. Midgley (eds.)). Cambridge University Press, Cambridge, United Kingdom and New York, NY, USA, 2013, 1535 pp.

Jacobson, M. Z.: Strong Radiative Heating Due to the Mixing State of Black Carbon in Atmospheric Aerosols. Nature, 409:695–697, 2001.


Kompalli, S.K., Babu, S.S., Moorthy, K.K., Manoj, M.R., Kirankumar, N.V.P., Shaeb, K.H.B., Joshi, A.K.: Aerosol black carbon characteristics over central India: temporal variation and its dependence on mixed layer height. Atmos. Res. 147–148, 27–37. http://dx.doi.org/10.1016/ j.atmosres.2014.04.015, 2014.

Kondo, Y., Matsui, H., Moteki, N., Sahu, L., Takegawa, N., Kajino, M., Zhao, Y., Cubison, M. J., Jimenez, J. L., Vay, S.,
Diskin, G. S., Anderson, B., Wisthaler, A., Mikoviny, T., Fuelberg, H. E., Blake, D. R., Huey, G., Weinheimer, A. J., Knapp, D. J., and Brune, W. H.: Emissions of black carbon, organic, and inorganic aerosols from biomass burning in North America and Asia in 2008, J. Geophys. Res., 116, D08204, doi:10.1029/2010JD015152, 2011.

Kumar, B., Chakraborty, A., Tripathi, S. N., & Bhattu, D.: Highly time resolved chemical characterization of submicron organic aerosols at a polluted urban location, Environ. Sci.: Process. Imp., 18(10), 1285-1296, 2016.

Laborde, M., Crippa, M., Tritscher, T., Jurányi, Z., Decarlo, P. F., Temime-Roussel, B., Marchand, N., Eckhardt, S., Stohl, A., Baltensperger, U., Prévôt, A. S. H., Weingartner, E., and Gysel, M.: Black carbon physical properties and mixing state in the European megacity Paris, Atmos. Chem. Phys., 13, 5831–5856, 2013.

Laborde, M., Mertes, P., Zieger, P., Dommen, J., Baltensperger, U., and Gysel, M.: Sensitivity of the Single Particle Soot Photometer to different black carbon types, Atmos. Meas. Tech., 5, 1031–1043, doi:10.5194/amt-5-1031-2012, 2012.

Lambe, A. T., Cappa, C. D., Massoli, P., Onasch, T. B., Forestieri, S. D., Martin, A. T., Cummings, M. J., Croasdale, D. R., Brune, W. H.,Worsnop, D. R., and Davidovits, P.: Relationship between oxidation level and optical properties of secondary organic aerosol, Environ. Sci. Technol., 47, 6349–6357, 2013.

Lawrence, M. G., and Lelieveld, J.: Atmospheric pollutant outflow from southern Asia: A review. Atmos. Chem. Phys., 10, 11,017 – 11,096, 2010.

Lee, S. H., Murphy,D. M., Thomson,D. S. ,Middlebrook,A.M.: Chemical components of single particles measured with Particle Analysis by Laser Mass Spectrometry (PALMS) during the Atlanta SuperSite Project: Focus on organic/sulfate, lead, soot, and mineral particles, J. Geophys. Res., 107(D1–D2), 4003,2002 doi:10.1029/2000JD000011.

Liu, D., Allan, J. D., Young, D. E., Coe, H., Beddows, D., Fleming, Z. L., Flynn, M. J., Gallagher, M. W., Harrison, R. M., Lee, J., Prevot, A. S. H., Taylor, J. W., Yin, J., Williams, P. I., and Zotter, P.: Size distribution, mixing state and source apportionment of black carbon aerosol in London during wintertime, Atmos. Chem. Phys., 14, 10061–10084, https://doi.org/10.5194/acp-14- 10061-2014, 2014.

Liu, D., Allan, J., Whitehead, J., Young, D., Flynn, M., Coe, H., McFiggans, G., Fleming, Z. L., and Bandy, B.: Ambient black carbon particle hygroscopic properties controlled by mixing state and composition, Atmos. Chem. Phys., 13, 2015–2029, doi:10.5194/acp-13-2015-2013, 2013.

Liu, D., Flynn, M., Gysel, M., Targino, A., Crawford, I., Bower, K., Choularton, T., Jurányi, Z., Steinbacher, M., Hüglin, C., Curtius, J., Kampus, M., Petzold, A.,Weingartner, E., Baltensperger, U., and Coe, H.: Single particle characterization of black carbon aerosols at a tropospheric alpine site in Switzerland, Atmos. Chem. Phys., 10, 7389–7407, doi:10.5194/acp-10-7389- 2010, 2010.



Mahapatra, P.S., Panda, S., Das, N., Rath, S., Das T.: Variation in black carbon mass concentration over an urban site in the eastern coastal plains of the Indian sub-continent, Theor. Appl. Climatol., 2013a, DOI 10.1007/s00704-013-0984-z.

Mahapatra,P.S., Ray,S., Das,N., Mohanty,A., Ramulu,T.S., Das,T., Chaudhury,G.R., Das,S. N.: Urban air-quality assessment and source apportionment studies for Bhubaneshwar, Odisha, Theor.Appl. Clim.,112,243-25, 2013b.

McMeeking, G. R., Morgan, W. T., Flynn, M., Highwood, E. J., Turnbull, K., Haywood, J., and Coe, H.: Black carbon aerosol mixing state, organic aerosols and aerosol optical properties over the United Kingdom, Atmos. Chem. Phys., 11, 9037–9052, 2011.

McMeeking, G. R., Hamburger, T., Liu, D., Flynn, M., Morgan, W. T., Northway, M., Highwood, E. J., Krejci, R., Allan, J. D., Minikin, A., and Coe, H.: Black carbon measurements in the boundary layer over western and northern Europe,
Atmos. Chem. Phys., 10, 9393–9414, 2010.

Middlebrook, A. M., Bahreini, R., Jimenez, J. L., and Canagaratna, M. R.: Evaluation of Composition-Dependent Collection Efficiencies for the Aerodyne Aerosol Mass Spectrometer using Field Data, Aerosol Sci. Tech., 46, 258–271, doi:10.1080/02786826.2011.620041, 2012.

Moffet, R.C. and Prather, K.A.: In-situ measurements of the mixing state and optical properties of soot with implications for
radiative forcing estimates, PNAS,106, 11872–11877, 2009.

Moorthy, K.K., Satheesh,S. K., Kotamarthi,V.R.: Evolution of aerosol research in India and the RAWEX–GVAX: an overview,Curr. Sci., 111,1,53-75, 2016,DOI: 10.18520/cs/v111/i1/53-75.

Moorthy, K.K.: South Asian aerosols in perspective: Preface to the special issue,Atmos. Environ, 125,307–311,2016, http://dx.doi.org/10.1016/j.atmosenv.2015.10.073.

Moorthy, K. K., Babu,S.S., Satheesh,S.K., Srinivasan,J. and Dutt, C.B.S.: Dust absorption over the "Great Indian Desert" inferred   using   ground-based   and   satellite   remote   sensing,   J.   Geophys.   Res.,   112,   D09206,2007 doi:10.1029/2006JD007690.

Moteki, N. and Kondo, Y.: Effects of mixing state on black carbon measurements by laser-induced incandescence, Aerosol Sci. Technol., 41, 398–417, 2007.

Moteki, N., Kondo,Y, Miyazaki,Y., Takegawa,N.,Komazaki,Y., Kurata,G., Shirai,T., Blake,D.R., Miyakawa,T., Koike,M: Evolution of mixing state of black carbon particles: Aircraft measurements over the western Pacific in March 2004, Geophys. Res. Lett., 34, L11803,2007, doi:10.1029/2006GL028943.

Moteki, N. and Kondo, Y.: Dependence of laser-induced incandescence on physical properties of black carbon aerosols: Measurements and theoretical interpretation, Aerosol Sci. Tech., 44, 663-675, 2010.

Miyakawa, T., Oshima,N., Taketani,F., Komazaki,Y., Yoshino,A., Takami,A., Kondo,Y., Kanaya,Y.: Alteration of the size distributions and mixing states of black carbon through transport in the boundary layer in east Asia, Atmos. Chem. Phys., 17, 5851–5864, 2017.

Ng , N. L., Herndon , S. C., Trimborn , A., Canagaratna, M. R., Croteau, P. L., Onasch, T. B., Sueper, D., Worsnop, D.R. , Zhang, Q. , Sun, Y. L., Jayne, J. T., 2011. An Aerosol Chemical Speciation Monitor (ACSM) for Routine Monitoring



of the Composition and Mass Concentrations of Ambient Aerosol, Aerosol Science and Technology, 45:7, 780-794, doi:10.1080/02786826.2011.560211.

Pandey, A., Sadavarte, P., Rao, A. B., Venkataraman, C.: Trends in multi-pollutant emissions from a technology-linked inventory for India: II. Residential, agricultural and informal industry sectors. Atmos. Environ., 99, 341-352, 2014..

Peng, J., Hu, M., Guo, S., Du, Z., Zheng, J., Shang, D., Levy Zamora, M., Zeng, L., Shao, M., Wu, Y.-S., Zheng, J., Wang, Y., Glen, C. R., Collins, D. R., Molina, M. J., and Zhang, R.: Markedly enhanced absorption and direct radiative forcing of black carbon under polluted urban environments, P. Natl. Acad. Sci. USA, 113, 4266–4271, doi:10.1073/pnas.1602310113, 2016.

Petzold, A., Ogren, J. A., Fiebig, M., Laj, P., Li, S.-M., Baltensperger, U., Holzer-Popp, T., Kinne, S., Pappalardo, G.,
Sugimoto, N., Wehrli, C., Wiedensohler, A., and Zhang, X.-Y.: Recommendations for reporting "black carbon" measurements, Atmos. Chem. Phys., 13, 8365–8379, doi:10.5194/acp-13-8365-2013, 2013.

Prasad, P., Ramana,R., Venkat Ratnam, M., Chen, W., Vijaya Bhaskara Rao, S., Gogoi, M.M., Kompalli, S.K., Kumar, K.S., Babu, S.S.; Characterization of atmospheric Black Carbon over a semi-urban site of Southeast India: Local sources and long-range transport, Atmos. Res., 213, 411–421, DOI:10.1016/j.atmosres.2018.06.024, 2018.

Raatikainen,T., Brus,D, Hooda1,R.K., Hyvärinen,A.P., Asmi,E., Sharma,V.P.,Arola,A., Lihavainen,H., Size-selected black carbon mass distributions and mixing state in polluted and clean environments of northern India, Atmos. Chem. Phys., 17, 371-383, doi:10.5194/acp-17-371-2017, 2017.

Raatikainen, T., Brus, D., Hyvärinen, A.-P., Svensson, J., Asmi, E., and Lihavainen, H.: Black carbon concentrations and mixing state in the Finnish Arctic, Atmos. Chem. Phys., 15, 10057– 10070, doi:10.5194/acp-15-10057-2015, 2015.

Reddington, C.L., McMeeking, G., Mann, G.W., Coe, H., Frontoso, M.G., Liu, D., Flynn, M., Spracklen, D.V., Carslaw, K.S.: The mass and number size distributions of black carbon aerosol over Europe. Atmos. Chem. Phys. 13, 4917– 4939, 2013.

Scarnato, B.V., China,S.,Nielsen, K., and Mazzoleni,C.:Perturbations of the optical properties of mineral dust particles by mixing with black carbon: a numerical simulation study,Atmos. Chem. Phys., 15, 6913–6928, 2015, www.atmos-chem-
phys.net/15/6913/2015/ doi:10.5194/acp-15-6913-2015.

Schwarz, J. P., Gao, R. S., Perring, A. E., Spackman, J. R., and Fahey, D. W.: Black carbon aerosol size in snow, Sci. Rep., 3, 1356, https://doi.org/10.1038/srep01356, 2013.

Schwarz, J. P., Gao, R. S., Spackman, J. R., Watts, L. A., Thomson, D. S., Fahey, D. W., Ryerson, T. B., Peischl, J., Holloway, J. S., Trainer, M., Frost, G. J., Baynard, T., Lack, D. A., de Gouw, J. A., Warneke, C., and Del Negro, L. A.:
Measurement of the mixing state, mass, and optical size of individual black carbon particles in urban and biomass burning emissions, Geophys. Res. Lett., 35, L13810, doi:10.1029/2008GL033968, 2008.

Schnaiter, M., Linke, C., Möhler, O., Naumann, K. H., Saathoff, H., Wagner, R., Schurath, U., and Wehner, B.: Absorption amplification of black carbon internally mixed with secondary organic aerosol, J. Geophys. Res., 110, D19204, https://doi.org/10.1029/ 2005JD006046, 2005.



Shiraiwa, M., Kondo, Y., Moteki, N., Takegawa, N., Miyazaki, Y., and Blake, D. R.: Evolution of mixing state of black carbon in polluted air from Tokyo, Geophys. Res. Lett., 34, L16803, https://doi.org/10.1029/2007GL029819, 2007.

Shiraiwa, M., Kondo, Y., Iwamoto, T., and Kita, K.: Amplification of light absorption of black carbon by organic coating, Aerosol Sci. Technol., 44, 46–54, 2010.

Srinivas, B., and Sarin, M.M.: PM2.5, EC and OC in atmospheric outflow from the Indo-Gangetic Plain: Temporal variability and aerosol organic carbon-to-organic mass conversion factor, Sci.of the Tot.Environ. 487,196–205,2014, http://dx.doi.org/10.1016/j.scitotenv.2014.04.002.

Srivastava, R. and Ramachandran, S.: The mixing state of aerosols over the Indo-Gangetic Plain and its impact on radiative forcing. Q. J. R. Meteorol. Soc. 139, 137–151. DOI:10.1002/qj.1958, 2013.

Takami, A., Mayama, N., Sakamoto, T., Ohishi, K., Irei, S., Yoshino, A., Hatakeyama, S., Murano, K., Sadanaga, Y., Bandow, H., Misawa, K., and Fujii, M.: Structural analysis of aerosol particles by microscopic observation using a time of flight secondary ion mass spectrometer, J. Geophys. Res. - Atmos., 118, 6726–6737, doi:10.1002/jgrd.50477, 2013.

Thamban,N.M., Tripathi,S.N., Shamjad P. M., Kuntamukkala,P., Kanawade, V.P.: Internally mixed black carbon in the Indo-Gangetic Plain and its effect on absorption enhancement. Atmos. Res., 197, 211–223, http://dx.doi.org/10.1016/
j.atmosres.2017.07.007, 2017.

Taylor, J. W., Allan, J. D., Liu,D., Flynn,M., Weber,R., Zhang,X., Lefer,B.L., Grossberg,N., Flynn,J., Coe, H.: Assessment of the sensitivity of core/shell parameters derived using the single particle soot photometer to density and refractive index. Atmos. Meas. Tech. Discuss., 7, 5491–5532, 2014.

Ueda,S., Nakayama,T., Taketani,F., Adachi,K., Matsuki,A., Iwamoto,Y., Sadanaga,Y., and Matsumi,Y.:Light absorption and
morphological properties of soot-containing aerosols observed at an East Asian outflow site, Noto Peninsula, Japan. Atmos. Chem. Phys., 16, 2525–2541, 2016.

Venkatraman, C., Habib, G., Eiguren-Fernandez, A., Mignel, A.H., Friedlander, S.K.: Residential biofuels in South Asia: carbonaceous aerosol emissions and climate impacts. Science 307:1454–1456, 2005.

Verma, S., Pani, S.K., Bhanja, S.N.: Sources and radiative effects of wintertime black carbon aerosols in an urban
atmosphere in east India. Chemosphere, 2012. doi:10.1016/j.chemosphere.2012.06.063.

Wang, Q. Y., Cao, J., Han,Y.,Tian, J., Zhu, C., Zhang, Y., Zhang, N., Shen, Z., Ni, H., Zhao, S., and Wu.,J.: Sources and physicochemical characteristics of black carbon aerosol from the southeastern Tibetan Plateau: internal mixing enhances light absorption. Atmos. Chem. Phys., 18, 4639–4656, 2018.

Wang, Q. Y., Huang, R. J., Zhao, Z. Z., Cao, J. J., Ni, H. Y., Tie, X. X., Zhao, S. Y., Su, X. L., Han, Y. M., Shen, Z. X.,
Wang, Y. C., Zhang, N. N., Zhou, Y. Q., and Corbin, J. C.: Physicochemical characteristics of black carbon aerosol and its radiative impact in a polluted urban area of China, J. Geophys. Res.-Atmos., 121,12505–12519, https://doi.org/10.1002/2016JD024748, 2016.


Wang, Q. Y., Huang, R.-J., Cao, J. J., Tie, X. X., Ni, H. Y., Zhou, Y. Q., Han, Y. M., Hu, T. F., Zhu, C. S., Feng, T., Li,N., and Li, J. D.: Black carbon aerosol in winter north-eastern Qinghai-Tibetan Plateau, China: the source, mixing state and optical property, Atmos. Chem. Phys., 15, 13059–13069, https://doi.org/10.5194/acp-15-13059-2015, 2015.

Weingartner, E., Burtscher, H., and Baltensperger, U.: Hygroscopic properties of carbon and diesel soot particles, Atmos. Environ.,31, 2311–2327, 1997.

Wu, Y., Zhang, R., Tian, P., Tao, J., Hsu, S.-C., Yan, P., Wang, Q., Cao, J., Zhang, X., Xia, X.: Effect of ambient humidity on the light absorption amplification of black carbon in Beijing during January 2013. Atmos. Environ. 124, 217–223, 2016.

Zhang, J., Liu, J., Tao, S., and Ban-Weiss, G. A.: Long-range transport of black carbon to the Pacific Ocean and its dependence on aging timescale, Atmos. Chem. Phys., 15, 11521–11535, https://doi.org/10.5194/acp-15-11521-2015, 2015.

Zuberi, B., Johnson, K. S., Aleks, G. K., Molina, L. T., and Molina, M. J.: Hydrophilic properties of aged soot, Geophys. Res. Lett., 32, L01807, https://doi.org/10.1029/2004GL021496, 2005.

**Figures and Tables**

**Figure 1:** Experimental location Bhubaneswar (star symbol); In the background, time averaged values of the aerosol optical depth at the wavelength 550 nm (color map) for the period 2009-2017 obtained using Moderate resolution Imaging Spectrometer (MODIS) (MODIS-Terra MOD08_M3 v6; combined dark target and deep blue product) are shown.

5   **Figure 2:** Isentropic five day airmass backtrajectories arriving at 100 meters above the surface over the observational location (identified with star symbol) in different seasons.

**Figure 3:** Temporal variation of daily mean (a) $r_{BC}$ mass concentration; and (b) number concentration of BC (bars) and non-BC scattering particles (filled circle). The vertical line passing through them is the standard deviation.

**Figure 4:** (a) Typical mass (number) size distributions along with least-squares fitting to mono- modal log-normal distribution (in dotted lines) used to derive MMD and CMD. (b) Temporal variation of daily mean mass medain diameter (triangle) and (b) Temporal variation of daily mean count median diameter (star) of BC; The symbols present the mean value for the day and the vertical line passing through them is the standard deviation. The solid continuous line shows the 30 day smoothed variation. Dotted vertical lines highlight different seasons.

**Figure 5:** Seaonal mean black carbon (a) mass size distributions and (b) numbe size distributions. Corresponding mode diameter values are also seen in brackets.

**Figure 6:** Temporal variation of daily mean relative coating thickness (hall filled  circle) and absolute coating thickness (star). The symbols present the mean value for the day and the vertical line passing through them is the standard deviation. The solid continuous line shows the 30 day smoothed variation. Dotted vertical lines highlight different seasons. Due to the failure of the scattering detector between 31-July-2016 to 20-September-2016 mixing state parameters could not be

20  estimated.

**Figure 7:** Frequency of occurrence of (a) relative coating thickness and (b) absolute coating thickness in different seasons.

**Figure 8:** Diurnal variation of (a-d) rBC mass concentrations and realtive coating thickness (RCT)  in different seasons. The vertical lines denote the Sunrise and Sunset. The vertical bars through solids points are the standard errors from the mean.

**Figure 9:** Seasonal variation of (a) mass concentrations and (b) percentage contributions to the total mass concentration of different species (organics, sulphates, nitrates, ammonium and chlorides)

**Figure 10:** Diurnal variation of mass fraction of different species (organics, sulphates, nitrates, ammonium and chlorides) of NR-PM1  in different seasons.

**Figure 11:** Association between mass-fraction of organics (top panels; a-d) and sulphates (bottom panels e-h) with relative coating thickness during different seasons. The colour bar indicates percentage of occurrence of RCT for corresponding MF

30  values of the species.

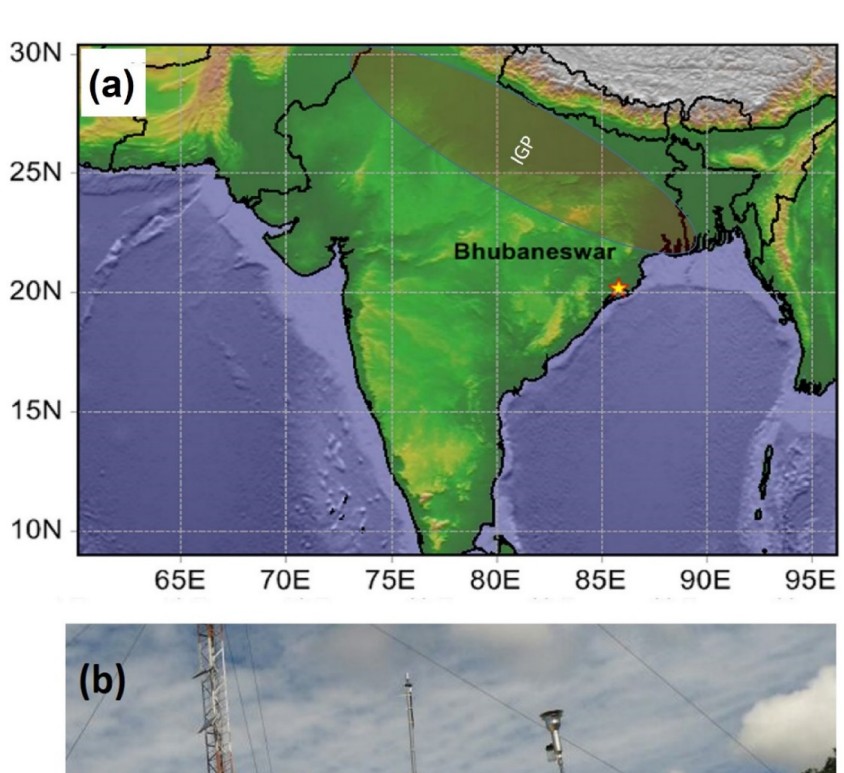

**Figure 1:** (a) Experimental location Bhubaneswar (star symbol); the Indo-Gangetic Plains (IGP) region is indicated with an oval symbol; (b) mobile container hosting the measurement setup at the experimental site.





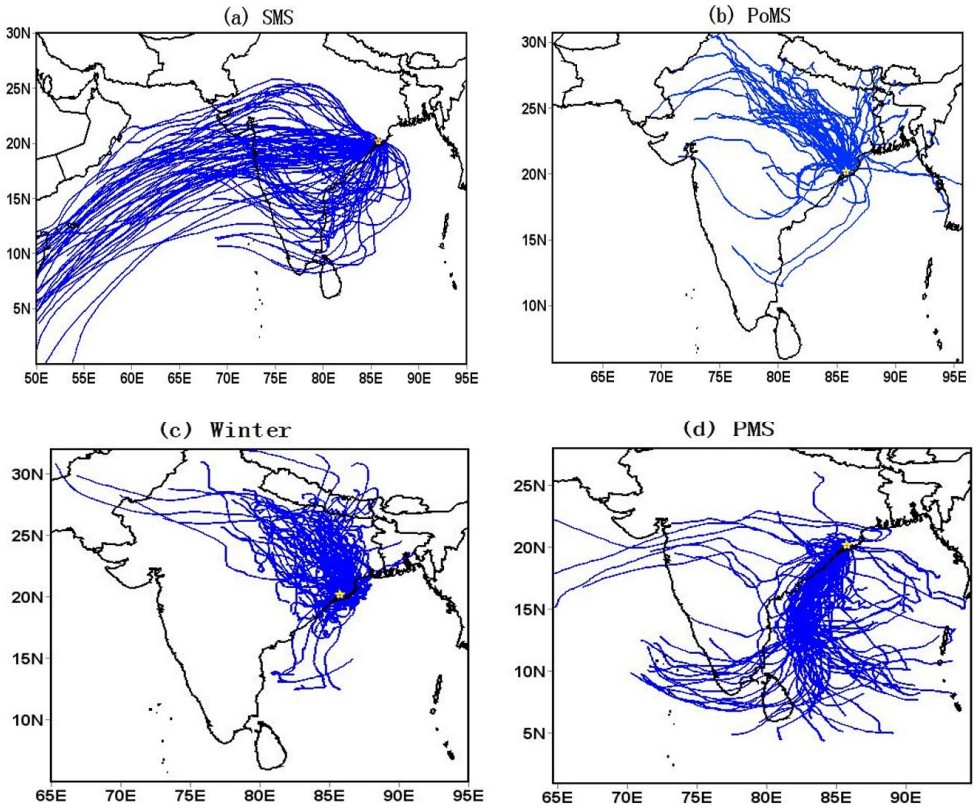

**Figure 2:** Isentropic five day airmass backtrajectories arriving at 100 meters above the surface over the observational location (identified with star symbol) in different seasons.

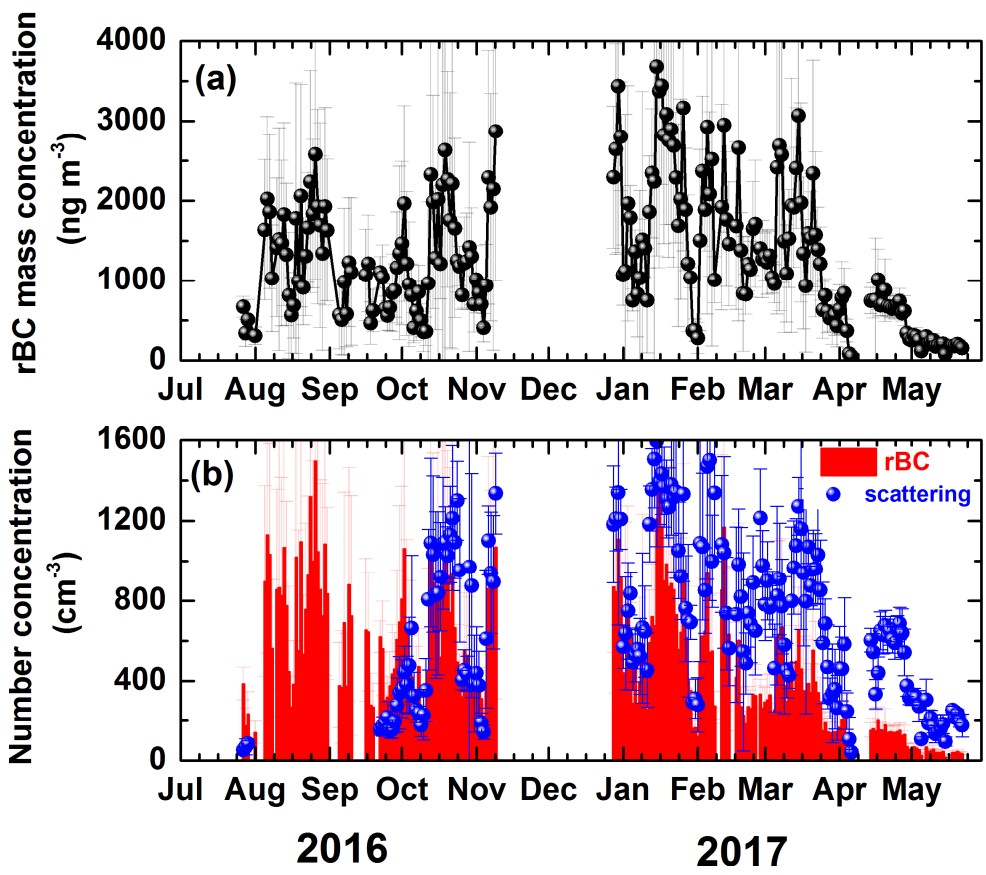

5 **Figure 3:** Temporal variation of daily mean (a) $r_{BC}$ mass concentration; and (b) number concentration of BC (bars) and non-BC scattering particles (filled circle). The vertical line passing through them is the standard deviation.



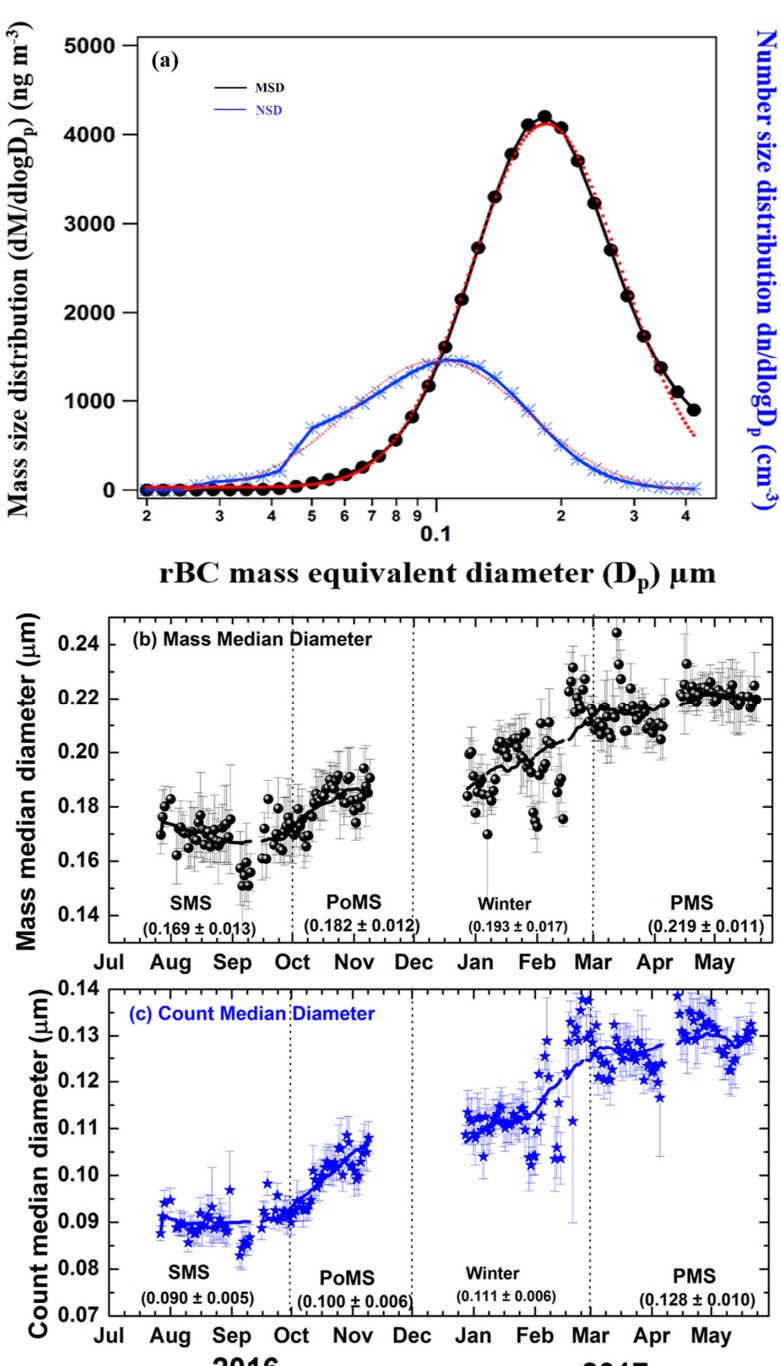





**Figure 4:** (a) Typical mass (number) size distributions along with least-squares fitting to mono- modal log-normal distribution (in dotted lines) used to derive MMD and CMD. (b) Temporal variation of daily mean mass medain diameter (triangle) and (b) Temporal variation of daily mean count median diameter (star) of BC; The symbols present the mean value for the day and the vertical line passing through them is the standard deviation. The solid continuous line shows the 30 day smoothed variation. Dotted vertical lines highlight different seasons.



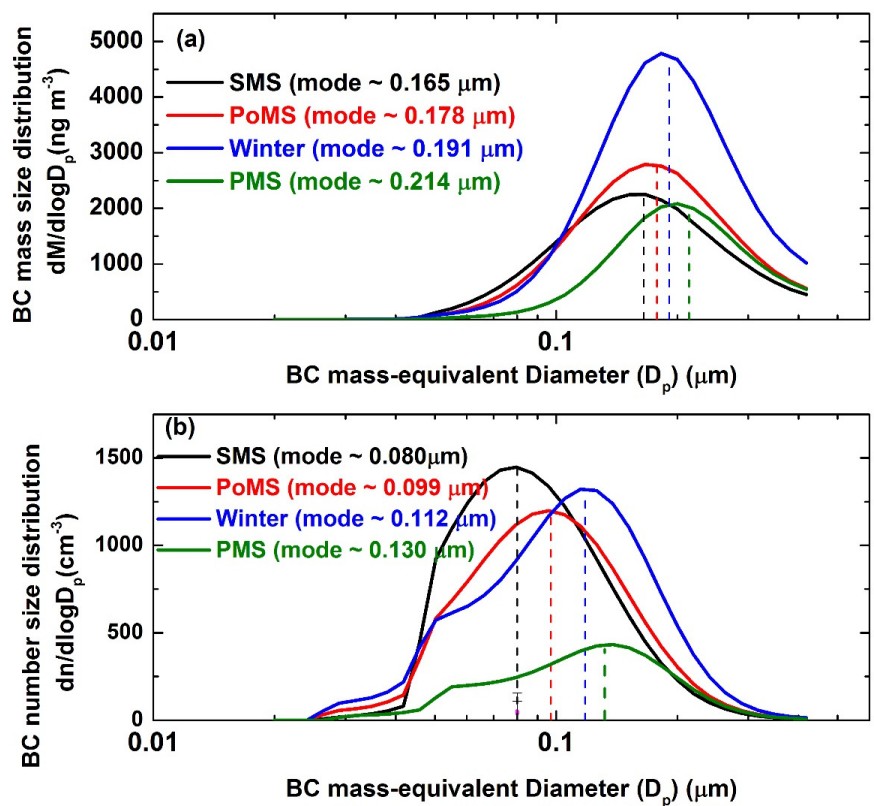

**Figure 5:** Seaonal mean black carbon (a) mass size distributions and (b) numbe size distributions. Corresponding mode diameter values are also seen in brackets.

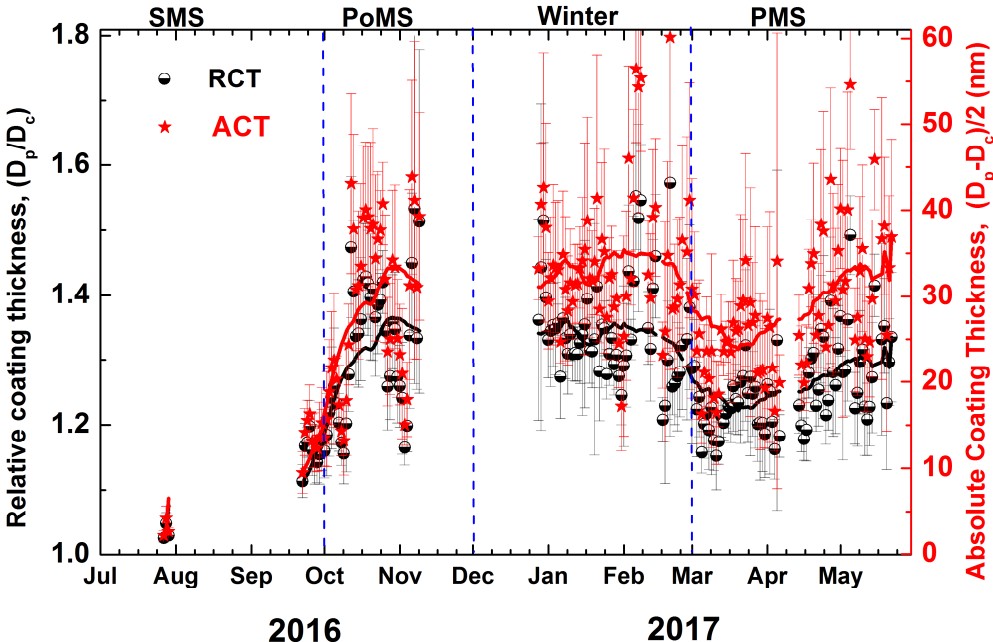

**Figure 6:** Temporal variation of daily mean relative coating thickness (hall filled circle) and absolute coating thickness (star). The symbols present the mean value for the day and the vertical line passing through them is the standard deviation. The solid continuous line shows the 30 day smoothed variation. Dotted vertical lines highlight different seasons. Due to the failure of the scattering detector between 31-July-2016 to 20-September-2016 mixing state parameters could not be estimated.

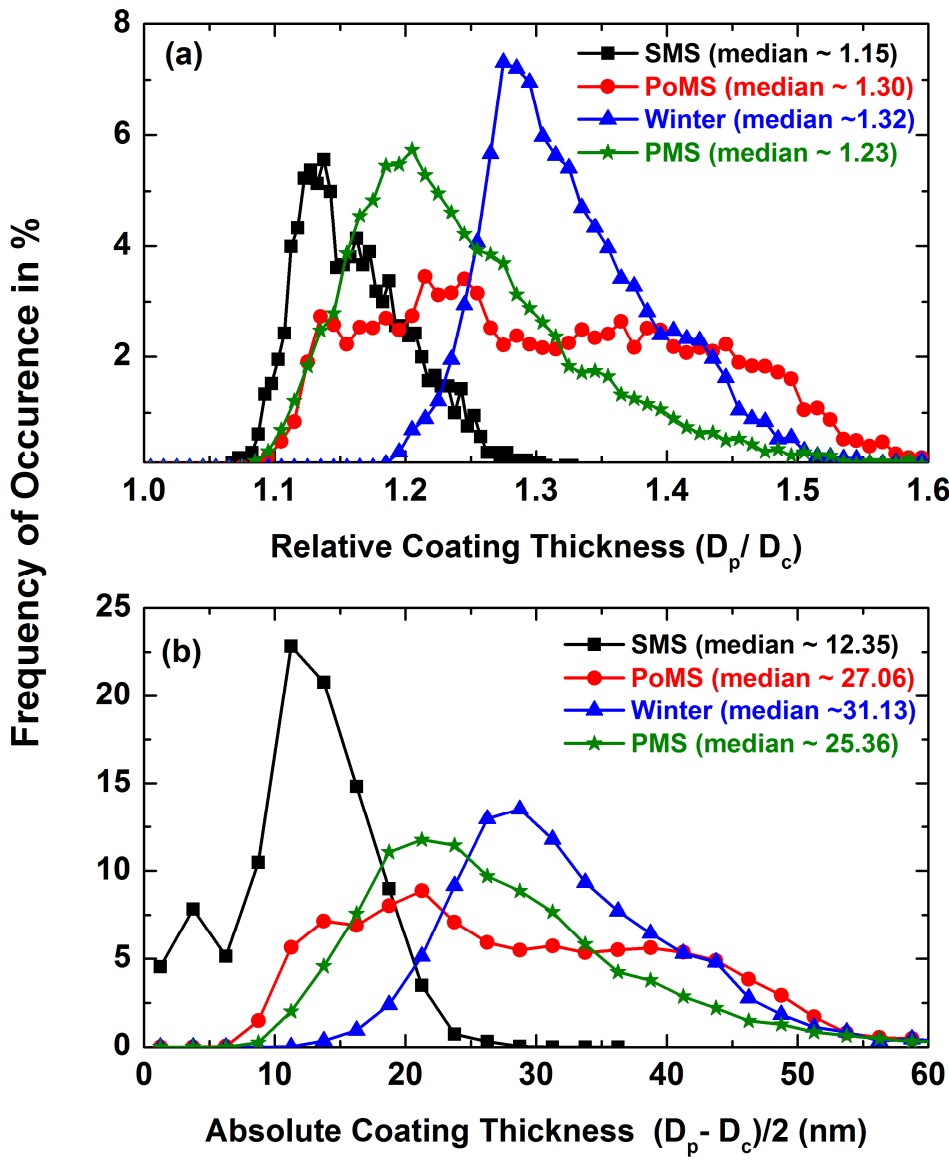

**Figure 7:** Frequency of occurrence of (a) relative coating thickness and (b) absolute coating thickness in different seasons.



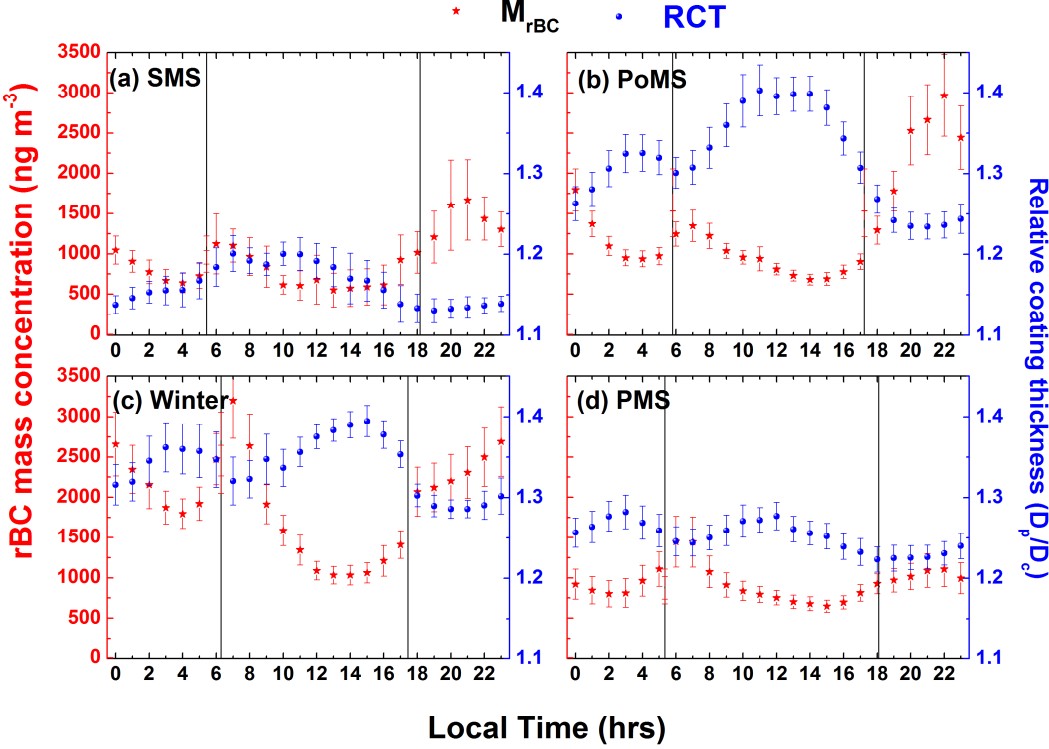

**Figure 8:** Diurnal variation of (a-d) rBC mass concentrations and realtive coating thickness (RCT) in different seasons. The vertical lines denote the Sunrise and Sunset. The vertical bars through solids points are the standard errors from the mean.





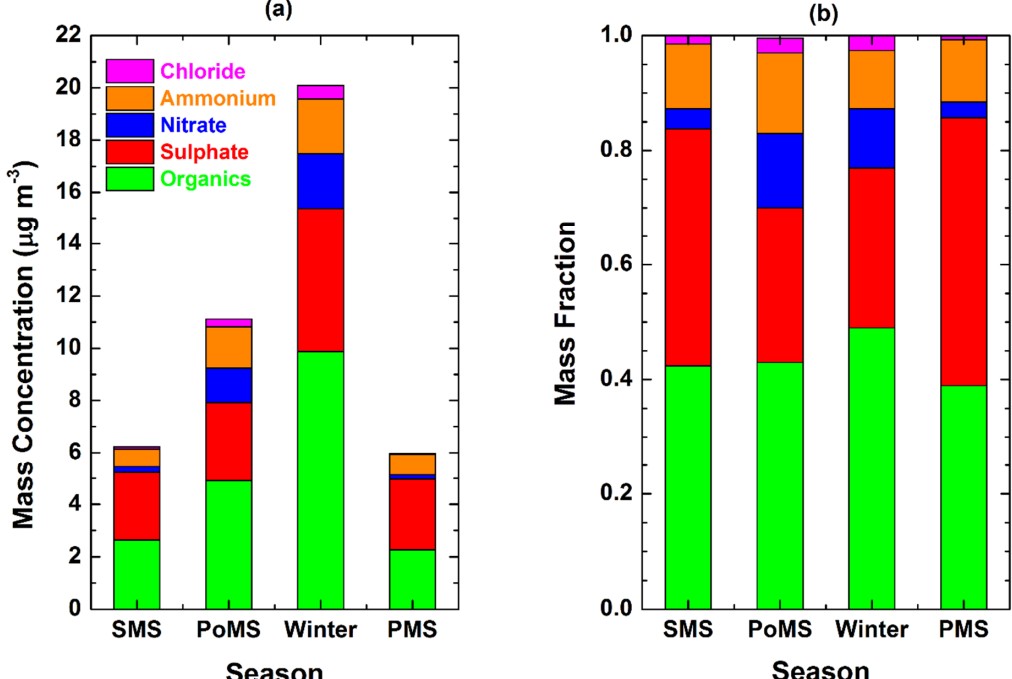

**Figure 9:** Seasonal variation of (a) mass concentrations and (b) percentage contributions to the total mass concentration of different species (organics, sulphate, nitrate, ammonium and chloride).



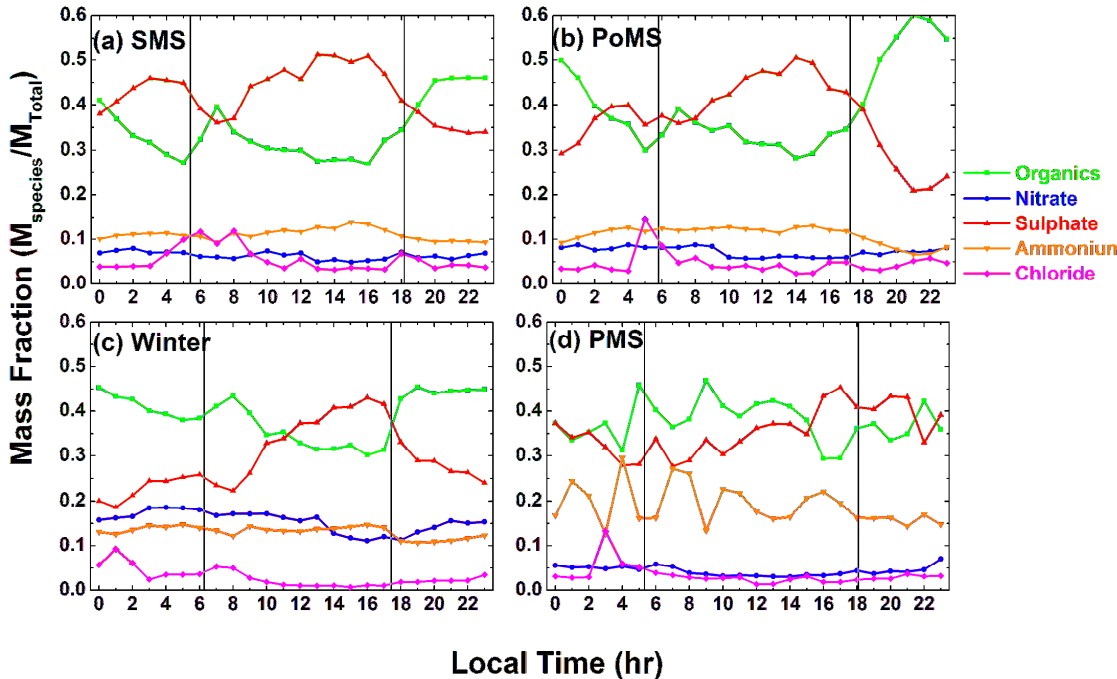

**Figure 10:** Diurnal variation of mass fraction of different species (organics, sulphate, nitrate, ammonium and chloride) of NR-PM1 in different seasons. The vertical lines denote the Sunrise and Sunset.





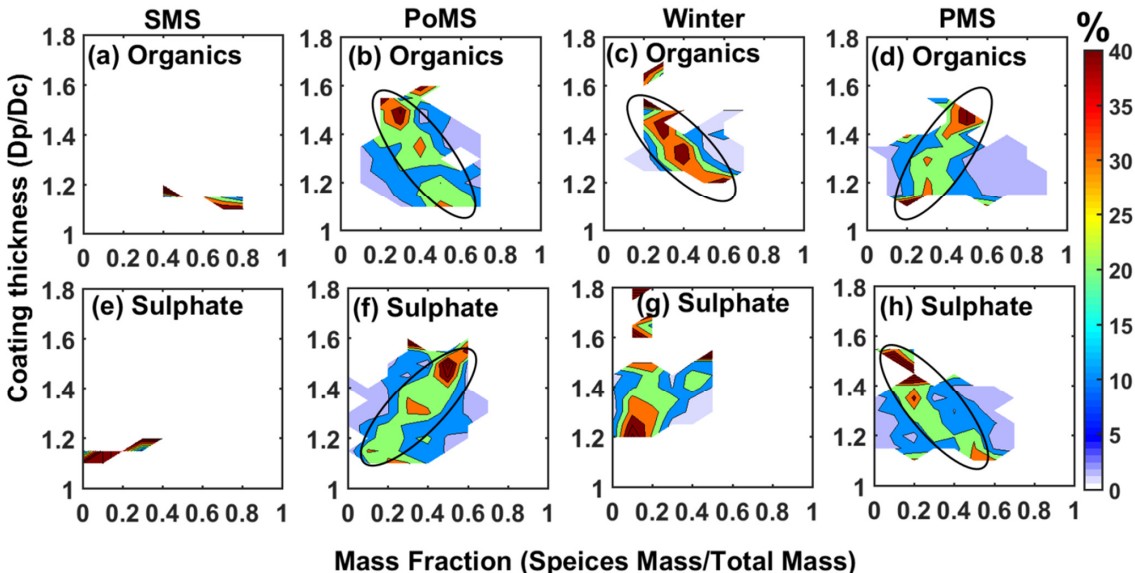

**Figure 11:** Association between mass-fraction of organics (top panels; a-d) and sulphate (bottom panels e-h) with relative coating thickness during different seasons. The colour bar indicates percentage of occurrence of RCT for corresponding MF values of the species.





**Table - 1:** Seasonal average of metrological parameters, temperature (T), Relative humidity (RH), Pressure (P) and Wind speed (WS). Maximum and minimum values recorded in that season are also listed.

| Seasons | | T ( °C ) | P ( hPa ) | WS ( m s$^{-1}$) | RH ( % ) |
|---|---|---|---|---|---|
| **SMS** | Mean | 30.12 ± 3.76 | 995 ± 3.6 | 1.23 ± 0.62 | 77.1 ± 17.5 |
| | Minimum | 23.9 | 986.2 | 0 | 31 |
| | Maximum | 35 | 1001.7 | 5.2 | 99 |
| **PoMS** | Mean | 28.3 ± 6.3 | 1002 ± 4.4 | 1.07 ± 0.67 | 71.0 ± 21.2 |
| | Minimum | 21 | 993.9 | 0 | 17 |
| | Maximum | 34 | 1010 | 4.8 | 99 |
| **Winter** | Mean | 25.5 ± 8.2 | 1006 ± 2.6 | 0.95 ± 0.53 | 57.4 ± 22.9 |
| | Minimum | 18 | 998.1 | 0 | 8 |
| | Maximum | 36 | 1001 | 3.4 | 99 |
| **PMS** | Mean | 33 ± 7.5 | 999.8 ± 3.6 | 1.97 ± 0.92 | 67.1 ± 28.2 |
| | Minimum | 18.9 | 987.7 | 0 | 7 |
| | Maximum | 41 | 1008.3 | 8 | 98.5 |

**Table 2:** Mean modal parameters (peak concentrations, mode and geometric standard deviation) of the BC mass and number size distributions considering the log-normal distribution in different seasons: summer monsoon (SMS), post-monsoon (PoMS), winter and pre-monsoon (PMS).

| (a) Mass size distribution | | | |
|---|---|---|---|
| **Season** | $M_0$ (ng m$^{-3}$) | Mode (nm) | Geometric standard deviation ($\sigma$) |
| **SMS** | 2225 | 0.165 | 1.99 |
| **PoMS** | 2752 | 0.178 | 1.90 |
| **Winter** | 4679 | 0.191 | 1.77 |
| **PMS** | 2031 | 0.214 | 1.72 |
| (b) Number size distribution | | | |
| **Season** | $N_0$ (cm$^{-3}$) | Mode (nm) | Geometric standard deviation ($\sigma$) |
| **SMS** | 1502 | 0.080 | 1.80 |
| **PoMS** | 1219 | 0.099 | 1.92 |
| **Winter** | 1270 | 0.112 | 2.01 |
| **PMS** | 407 | 0.130 | 2.01 |





**Table 3.** A Summary of BC mass median diameter from few selected studies representing different sources in distinct environments.

| S.No. | Location | Type of location | MSD mode/ MMD (µm) | Reference |
|---|---|---|---|---|
| **Urban/suburban locations** | | | | |
| *1.* | *Bhubaneswar, India* | *Urban/continental outflow* | *0.165-0.214* | *Present study* |
| 2. | Canadian oil sand mining (Aircraft studies), Canada | Urban/fresh urban emissions | 0.135-0.145 | Cheng et al. (2018) |
| 3. | Gual Pahari, India | Urban polluted/ fresh biofuel, crop residue | 0.221 ± 0.014 | Raatikainen et al. (2017) |
| 4. | Shangai, China | Urban/pollution episode with high biomass burning | 0.230 | Gong et al. (2016) |
| 5. | Suzu, Japan | Urban/east Asian out flow site | 0.200 | Ueda et al. (2016) |
| 6. | An urban site in London, UK | Urban/traffic emissions | 0.119-0.124 | Liu et al. (2014) |
| 7. | Suburban site in Paris, France | Urban/traffic emissions | 0.100-0.140 | Laborde et al. (2013) |
| 8. | Sacramento, USA | Urban/fossil fuel emissions | ~0.145 | Cappa et al. (2012) |
| 9. | Tokyo, Japan | Urban outflow | 0.130-0.170 | Kondo et al. (2011) |
| 10. | Cranfield airport in UK | Aircraft emissions near source | 0.126 | McMeeking et al. (2010) |
| 11. | Regionally-averaged over flight segments over Europe | Near source to free troposphere | 0.170-0.210 (a) continental pollution (0.18–0.21); (b) urban outflow (0.170±0.010) | McMeeking et al. (2010) |
| **Remote locations** | | | | |
| 12. | Lulang, Tibetan Plateau, China | High-altitude background | 0.160 ± 0.023 | Wang et al. (2018) |
| 13. | Mukteshwar, The Himalayas, India | High-altitude background / biofuel, crop residue outflow | 0.205 ± 0.016 | Raatikainen et al. (2017) |
| 14. | Northeastern Qinghai–Tibetan Plateau, China | Background site/biomass burning, aged BC | 0.187 | Wang et al. (2015) |
| 15. | Jungfraujoch, Switzerland | High-altitude background / biomass burning, aged BC | 0.220-0.240 | Liu et al. (2010) |





**Table 4**: A summary of the properties of the rBC concentrations, size distributions and its mixing state and scattering particle concentrations in different seasons: summer monsoon (SMS), post-monsoon (PoMS), winter and pre-monsoon (PMS). The values after ± symbol are standard deviations.

| Parameter | SMS | PoMS | Winter | PMS |
|---|---|---|---|---|
| BC mass concentration (ng m$^{-3}$) | 941 ± 615 | 1338 ± 1396 | 1935 ± 1578 | 816 ± 835 |
| BC number concentration (cm$^{-3}$) | 500 ± 322 | 583 ± 616 | 621 ± 557 | 188 ± 196 |
| Scattering particle concentration (cm$^{-3}$) | 211 ± 114 | 690 ± 471 | 950 ± 464 | 548 ± 349 |
| Mass median diameter (μm) | 0.169 ± 0.013 | 0.182 ± 0.012 | 0.193 ± 0.017 | 0.219 ± 0.011 |
| count median diameter (μm) | 0.090 ± 0.005 | 0.100 ± 0.006 | 0.111 ± 0.006 | 0.128 ± 0.010 |
| Relative coating thickness | 1.16 ± 0.04 | 1.32 ± 0.14 | 1.34 ± 0.12 | 1.26 ± 0.10 |
| Absolute coating thickness (nm) | 24.24 ± 9.9 | 56.94 ± 23.76 | 65.01 ± 15.80 | 55.02 ± 19.25 |

**Appendix**

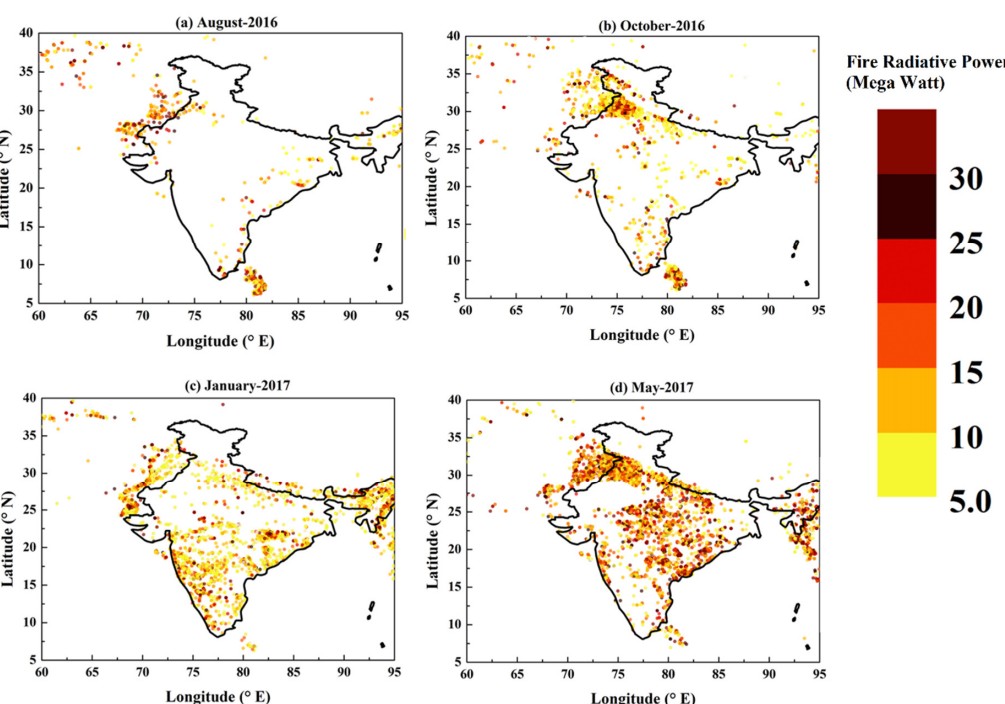

**Figure A1:** Spatial distribution of Moderate Resolution Imaging Spectroradiometer (MODIS) fire radiative power (MODIS
Thermal Anomalies / Fire locations Collection 6 product obtained from https://earthdata.nasa.gov/firms) for the
representative months of different seasons; (a) August -2016 (SMS), (b) October -2016 (PoMS), (c) January -2017
10    (winter)  and (d) May -2017 (PMS). Significant amount of fire events during PMS are clearly seen over the Indian
region. During the PoMS (northwest IGP) and winter (western, north eastern regions of India) less intense regional fire
events are noticeable.

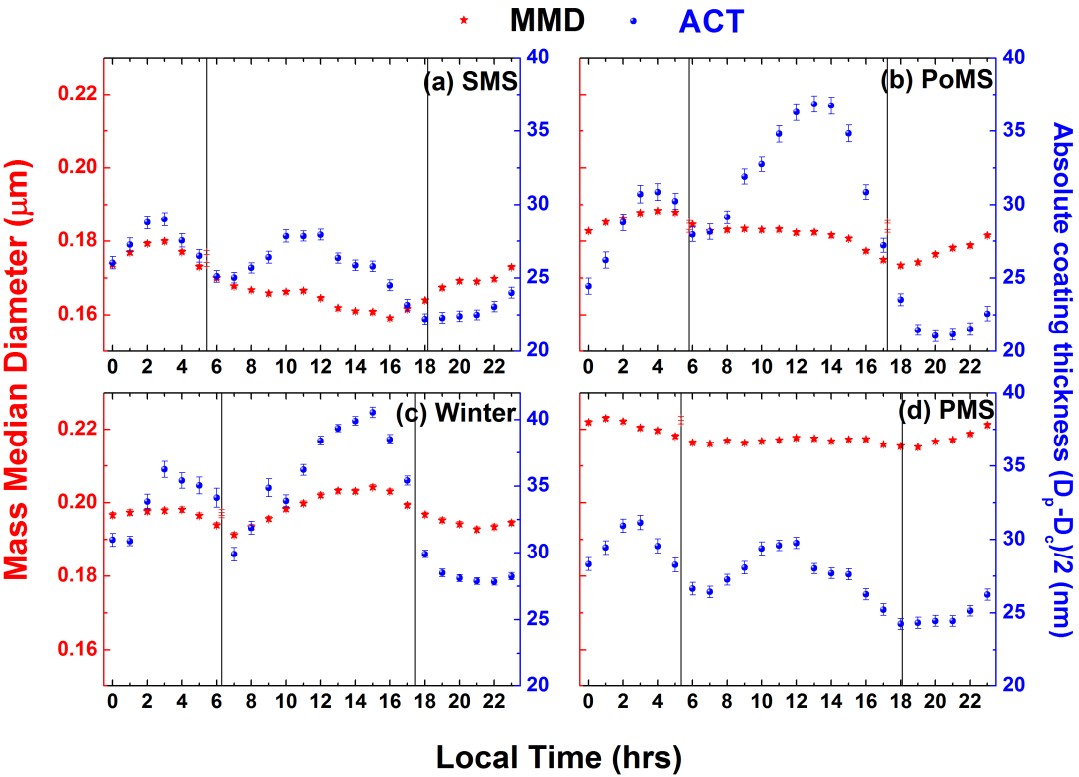

5 **Figure A2:** Diurnal variation of (a-d) rBC mass median diameter and absolute coating thickness (ACT) in different seasons. The vertical lines denote the Sunrise and Sunset. The vertical bars through solids points are the standard errors from the mean.