# Peer review of "Seasonal contrast in size distributions and mixing state of black carbon and its association with PM1.0 chemical composition from the eastern coast of India"

_Atmospheric Chemistry and Physics, 2019_

## Referee Comment (RC1) · Anonymous Referee #1 · 16 Jul 2019

The size of BC clearly shows seasonal variation and emission sources of BC. They selected a suitable sampling location that gets plumes from both lands as well as from coastal/marine regions on different seasons. They used SP2 and ACMS to characterize the microphysical and chemical properties of BC and other aerosol particles. The research provides useful information on BC mixing with other aerosol components based on a diurnal and seasonal variation on the eastern coast of India. IGP is one of the regional hot spots for BC aerosol concentration in south Asia and the present research provides detail information on the mixing scenario of BC aerosol from the IGP

and compares with the BC from other parts of India. They quantified the coating of BC in term of ACT and RCT and presented the influence of different coating materials and the discussed the preferential coating in different seasons. The language used is good but can be polished, especially, long sentences are used, and it makes difficult to communicate the message. I recommend the authors to shorten/split the long sentences. I recommend the publication of this article but after addressing the concerns listed below. How relevant is the discussion of mineral dust in the introduction section (paragraphs 1 and 2)? I understand that the authors highlight the light absorbing nature of minerals. But, I did not find any further discussion on minerals in the manuscript. In that context, the direct focus on BC mixing state is meaningful. The authors mention the use of 2 kinds of aerosol characterization instruments during the study. But there is no mention of one of the two instruments throughout the introduction section. I had to go through the method section to get information about the second instrument. Please mention a brief introduction on the second instrument used in this study. Regarding the use of SP2, a recent finding by Sedlacek III et al. (2018) cautioned about the charring of organic depending on the SP2 laser power (Sedlacek III, Arthur J., et al. "Formation of refractory black carbon by SP2-induced charring of organic aerosol." Aerosol Science and Technology 52.12 (2018): 1345-1350). It is worthy to mention this caveat as the ambient aerosol samples include organics. Clearly mention the power of the laser used during the operation of the SP2. Also, it is not always appropriate to assume core-shell structure for coated BC due to the complex mixing state of BC such as the case when BC is located off-center. Though the study did not utilize single particle off-line analysis to probe the complex internal mixing state of BC, it is useful to mention the effects from such BC structures on absorption and scattering signals (e.g., Sedlacek III, Arthur J., et al. "Determination of and evidence for non‐core‐shell structure of particles containing black carbon using the Single‐Particle Soot Photometer (SP2)." Geophysical research letters 39.6 (2012)). The authors should discuss how such non-core-shell particles would affect their results. For all instruments employed, also mention the model number in addition to the manufacturing company. Be consistent in the

use of units such as for flow rate and BC concentration. The use of different units for the same quantity makes it difficult to infer the values. Minimize the use of numeral values for comparison all the times. It may appear to the authors that these numerals are useful for comparison, but to me, it is a source of distraction. I recommend listing the values in Tables which the authors have already done. Authors can infer Table for values and focus on their findings. General comments: Page4 line 19: Add some references on the size of monomers for nascent BC. Cite (2017): 166, Köylü, Ümit Özgür, et al. "Fractal and projected structure properties of soot aggregates." Combustion and Flame 100.4 (1995): 621-633; Bhandari, Janarjan, et al. "Effect of thermodenuding on the structure of nascent flame soot aggregates." Atmosphere 8.9. Page4 line19: Also, it is not correct to say that the coated BC is 'spherical' after coating. Rather I would prefer to use the term like 'compact' or 'collapsed' for coated BC core, though the core-shell model treats such coated BC as the spherical core for simplicity. Page 4 line 27: What are you referring to using "these"? I am not clear. Page4 line 32: What sources? Do you mean 'aerosol sources'? Be specific Page 5 line 24: Only one instrument is revealed. But the authors mention in the abstract that they used 2 instruments. What is the second instrument used? Specify the second instrument as well in the introduction section mentioning why the instrument was selected. Page 5 line 25: Do the authors mean to say that the working of SP2 is based on long-term measurements only? Please rephrase the sentence. Page 6 line 12: Do you mean to say 'above ground level' by acronym AGL? Please mention the full name for the first time. Page 6 line 28: By 'Supplementary Figure S1' are you referring for the figures in an appendix? If so rename the figure as A1. Page 7 lines 1-8: As the numeral values for meteorological parameters are shown in Table 1, avoid using all these numerals in the text for comparison. Please minimize the use of numeral values in the text. Include only those specific values that are striking to discuss. Page 7 lines 15-17: Include the model number and company name for each instrument Page8 line2: It is mentioned that the RI of 2.26 – 1.26i is used for BC. Is this RI representative for ambient BC aerosol in the region? I am aware of the use of the above-mentioned RI value for BC.

However, the RI of 2.26 – 1.26i looks higher than usually used value for the RI of BC. Page8 line 9: As mentioned earlier, be consistent in picking unit for a given quantity. Here for flow rate, you used cm3/min while in line 21 you used liters/minute. Page 8 line 15: For the 40-100 nm range, mention clearly that the size is 'aerodynamic diameter'. Page 8 line 26: The sub-heading 3.1 can be made more specific. By 'Mass and number concentration' only it is not clear what is being measured. Pages 8-9 section 3.1: As mentioned earlier, use as least numeral values as possible in the discussion. All the numerals can be summarized in the table. Also, use the same units for particle concentration. In some cases, ng/m3 is used in some instance, $\mu$g/m3 is used for particle concentration. Page 10 line 27: I am not clear about this '...reported for reported from...'. Please clarify this sentence. Page 11 line 2: Is 'Figure S1' the same labeled as 'Figure A1' in the appendix? Add label for each season in the map. In fig. A1 (a) and (b), it will be useful to mention the location of the distinct data points below south India as a note. Page 11 line 10: It is mentioned that the BC mass loading was lowest during PMS, but fire events were maximum during PMS throughout the Indian region as shown in Fig. A (d). Is not it reasonable to expect a high concentration of BC? Page 14 lines7-9: The sentence "The figure reveals…." is not clear to me. Please rewrite the sentence Page 14 line11, line 13: Do you mean to say that the RCT of BC is contributed by the day-night temperature difference in different seasons? I am not clear. Page 15 lines6-9: The sentence 'Interestingly…occurring' is a long sentence. Page 15 lines 13-16: The sentence 'Not…role' is again a long sentence and is not clear. Simplify your statement. Page 16 line 28: Chlorine is shown to be present in very low concentration even when air mass arrived from marine region to the sampling site during PMS. Is this concentration normal during the PMS as well? Page 17 lines10-13: The sentence 'Further…. mixed' is not clear. In "absorption condensable species", do you mean 'absorption of condensable species' and the part "which already more internally mixed" is not clear.

Technical comments: Page4 line 16: Correct the year for 'China et al., 2012' to make it '2013'. Page4 line 27: A sentence starts with a pronoun at the beginning of a paragraph. It makes it difficult to know what you are referring to using a pronoun in the very beginning of a paragraph. Page 10 line3: What is represented by 'A' in equation (1)? Mention it. Page 11 line1: add 'of' before 'larger-sized BC particles. . .'. Page 20 line 2: Stronger is already a comparative adjective. Remove 'more' before stronger.
* * *

---

## Referee Comment (RC2) · Anonymous Referee #2 · 4 Nov 2019

This paper describes measurements of black carbon (BC) containing particles and of non-refractory PM1 over 9-10 months at an urban site on the east coast of India. The data show some interesting seasonal trends and are worth publishing. However, the paper itself needs significant work. There are many missing words, misused words, confusing sentences and repetitive sections. It should not be such hard work to figure out what the authors are trying to say. Since the paper has several UK co-authors, I am surprised that it was not carefully edited by a native English speaker before submission. This paper needs major revisions before it is accepted for publication.

Specific comments:

1- The introduction is rather repetitive and does not mention the second instrument, the ACSM. Please tighten up the language.

2- page 2, lines 17 -25. It would be helpful to mention which months are included in each season.

3- page 2, lines 28 - 33: The statement "The diurnal pattern of sulphate resembled that of the RCT" and the statement "the coating on BC showed a negative association with sulphate" contradict each other. The association plot in Figure 11 is an interesting way to present the data, but the conclusions are tenuous and don't belong in the abstract. I would delete the sentence starting "Though the pre-monsoon. . ." and ending ". . .mixing state of BC."

4- page 4, line 6: VOCs are not an aerosol species.

5- page 4, line 19: A diameter of a few tens of nanometers is more likely for the primary spherules than for the chain agglomerates. Please cite a reference or correct the text.

6- page 4, line 27: Paragraph starts with the word "These," but it is not clear what "these" is referring to.

7- page 6, line 9: Figure 1b can be removed. No-one needs to see a picture of a shipping container.

8- page 6, line 11: Example of poor English usage "away from the proximity of." Just say "not near."

9- page 6, line 25: You refer to Figure S1, but the extra figures are in an appendix. Please label extra material consistently.

10- page 6, line 30: I don't understand what the words in parentheses are conveying. If you mean that there are more fire events in those regions, please rephrase the sentence.

11- page 7, lines 10-14: What about the 6-week gap in November-December? That is more than a brief gap and needs some explanation.

12- page 7, lines 17-18: Why mention instruments that are not relevant to this study?

13- page 7, lines 31 – page 8, line 4: What do you mean by scattering enhancement? I think you are deriving the optical diameter of the coated particle from the scattering signal using a Mie Scattering model, but this description is garbled. Please phrase this more clearly.

14- page 8, line 15: Specify that this size range is vacuum aerodynamic diameter. The range is really more like 80 nm – 800 nm.

15- page 8, lines 19-20: It really doesn't seem necessary to discuss the pumps on the ACSM.

16 – page 10, lines 6-7 and elsewhere: Please pick one term, either counts or number, and use it consistently throughout the paper including the figures.

17- page 10, first paragraph: Why is there a distinct jog at 40 nm in the number size distributions?

18- page 16, line 16: What does "evolving least-squares fitting" mean?

19- pages 10-11: I find this discussion extremely confusing and repetitive. The most important data is displayed in Figures 3 and 4 (and Table 4). I do not understand the point of averaging the size and number distributions over a season, taking the mode and getting a slightly different number than the average of the mode for individual data points. This does not add any new information and leads to repetitive discussion of the results. I would remove Figure 5, Table 2 and the associated discussion. Similarly, with the peak of the seasonal number size distributions – what new information does this give you beyond what you already know from the BC mass loading? The discussion about previous work is split between page 10, lines 18-22 and page 11, line 29 - page 12 line 4. Please consolidate. Finally, if you think you can tell the difference between

local emissions during SMS with a smaller size and continental outflow with a larger size, why not make two entries in Table 3 for this study?

20- Pages 12-13: The discussion of RCT and ACT is confusing and repetitive. You are making a major assumption of core-shell morphology in order to calculate Dp and therefore RCT and ACT. Really all you can say is that you have a ratio that represents the amount of non-BC material associated with BC – you don't know the morphology of the particles nor how it changes with season. Morphology is likely to be quite different between fresh emissions during SMS and aged emissions during other seasons. I would not interpret RCT and ACT as literal diameters and coating thicknesses. In fact, I do not think ACT adds to the discussion. I would rewrite this section to present only the RCT data and include enough caveats that it is clear RCT is a representation, not an actual ratio of diameters. I also don't understand the point of Figure 7. You already have the information about the width of the distributions in Table 4. You mention multiple maxima, but have no interpretation. The discussion of Figure 7 on page 13 repeats the same information about sources and processing as on page 12, making this section very repetitive. I would remove Figure 7 and the associated discussion.

21- Page 12, lines12-14: "Both of these parameters...mixing state of the particles." I do not understand this sentence. Does "both" refer to RCT and ACT or to Dp and Dc? Either way, how can Dp not depend on the mixing state?

22- Page 13, lines 1-2: "Intra-seasonal variability...Figure 6) is also higher during PoMS." I do not understand this sentence. Figure 6 shows daily values, not seasonal values. By eye, the variation in the daily points and the spread of the error bars looks very similar across PoMS, Winter and PMS. There are very few points during SMS, so it is hard to draw conclusions for that season.

23- Pages 15-17, Section 3.5: I have several questions about the ACSM data analysis. Why is ammonia so low? Was the aerosol not neutralized and do you have corroborating evidence? Or was the RIE_NH4 incorrect for this instrument? Why not estimate

OOA and HOA using the parameterization in Ng et al. (EST 2011)? This would give you additional information about local and regional sources.

24- Page 17, lines 1-7: "Even though...coating." I do not understand this sentence, partly because it is too long and convoluted, but also because the two parts contradict each other. You say in part a that concurrent peaks in RCT and sulphate suggest that sulphate is mixed with BC, but in part b, you say the opposite. You can't have it both ways. Or are you saying that the ACSM detects sulphate when it is mixed with BC, but not organic? That does not make sense.

25- Pages 17-18, Section3.7: Figure 11 is another way of comparing the diurnals for RCT and MF. While it is a nice visualization, I don't think it needs a new section repeating much of the discussion as in Section 3.6. I would combine the discussion in Sections 3.6 and 3.7. I also wonder if you have thought about the fraction of particles containing BC (i.e., BC number con./(BC num conc. + scatt num conc.))? This fraction is much higher in SMS than in PoMS or Winter and is lowest in PMS. The low value in PMS might be part of the reason that the association in Figure 11d and h is so poor since there is less overlap between the particle population detected with the ACSM and the population detected with the SP2.

26- Figure 1: It is very hard to see the circle indicating the IGP in panel (a). I would delete panel (b). There is no need for a picture of a shipping container.

27- Figure 2: The star symbol is not visible.

28- Figure 3: Indicate the seasons in panels (a) and (b).

29- Figure 4: Include the dashed lines in the legend. Panel (b) has circles not triangles. Please use either number or count, but not both. Do you have any data covering the gap between end of May and August? Do the MMD and CMD really drop from PMS to SMS values over 6 weeks? Or could you have some kind of instrumental drift that causes both to increase over the displayed 10 months of data? 30- Figure 11:

"Speices" is mis-spelled in the x-axis label.

31- Table 1: "metrological" should be "meteorological"

32- Table 3: "Shangai" is mis-spelled.

33- Table 4: Are these averages of the daily values shown in the figures, or averages of all the underlying data? Please specify. Also, somewhere in the text you should state the time-base of the SP2 data.

34- Figure A1 (or S1?): Please decide if this section is an appendix or supplemental information. Please label the panels with the season. It's not clear what the words in parentheses in the last sentence are supposed to mean.

---

## Author Comment (AC1) · 15 Dec 2019

**Response to reviewers**

**Observations of the reviewers are in italics, and our response is given below in bold letters.**

**Reviewer-1: ACP-2019-376-RC1**

*The size of BC clearly shows seasonal variation and emission sources of BC. They selected a suitable sampling location that gets plumes from both lands as well as from coastal/marine regions on different seasons. They used SP2 and ACMS to characterise the microphysical and chemical properties of BC and other aerosol particles. The research provides useful information on BC mixing with other aerosol components based on a diurnal and seasonal variation on the eastern coast of India. IGP is one of the regional hot spots for BC aerosol concentration in south Asia and the present research provides detail information on the mixing scenario of BC aerosol from the IGP and compares with the BC from other parts of India. They quantified the coating of BC in term of ACT and RCT and presented the influence of different coating materials, and they discussed the preferential coating in different seasons. The language used is good but can be polished, especially, long sentences are used, and it makes difficult to communicate the message. I recommend the authors to shorten/split the long sentences. I recommend the publication of this article but after addressing the concerns listed below.*

**We thank the reviewer for the summary comments and positive recommendation, followed by detailed evaluation, and constructive comments. We have revised the manuscript accordingly, and our responses to the specific comments of the reviewer are given below:**

*How relevant is the discussion of mineral dust in the introduction section (paragraphs 1 and 2)? I understand that the authors highlight the light-absorbing nature of minerals. But, I did not find any further discussion on minerals in the manuscript. In that context, the direct focus on BC mixing state is meaningful.*

**Complied with. This was originally included since dust constitutes a major fraction of aerosols loading over the IGP during spring and summer. However, we agree that dust is not the theme of this paper, and so we have modified the introduction section to focus mainly on the BC mixing state.**

*The authors mention the use of 2 kinds of aerosol characterization instruments during the study. But there is no mention of one of the two instruments throughout the introduction section.*

*I had to go through the method section to get information about the second instrument. Please mention a brief introduction on the second instrument used in this study.*

**Complied with. We have included the relevant details in the introduction section. The following sentences are included in the revised manuscript.**

**Page 5, Line 14: "Along with this, information on the condensable materials which act as coating substances and constantly alter the physiochemical properties of the BC containing particles, is also essential. Collocated mass spectroscopy-based high-resolution aerosol chemical composition measurements have been employed for this purpose (Liu et al., 2014; Gong et al., 2016)."**

**Page 5, Line 25: "To meet these objectives, state-of-the the-art instruments were installed at Bhubaneswar, which included a single particle soot photometer (SP2) for characterization of refractory BC (rBC) aerosols and an Aerosol Chemical Speciation Monitor (ACSM) for high-resolution measurements of non-refractive submicron aerosol chemical composition for long-term measurements."**

**Page 5: Line 31: "The contributions from distinct sources to BC concentrations and the association of coating on BC with possible condensable coating materials are examined, and the implications are discussed."**

*Regarding the use of SP2, a recent finding by Sedlacek III et al. (2018) cautioned about the charring of organic depending on the SP2 laser power (Sedlacek III, Arthur J., et al. "Formation of refractory black carbon by SP2-induced charring of organic aerosol." Aerosol Science and Technology 52.12 (2018): 1345-1350). It is worthy to mention this caveat as the ambient aerosol samples include organics. Clearly mention the power of the laser used during the operation of the SP2.*

**Thank you, and we agree. We have included this in the revised manuscript. Page 8, Line24.**

**"Recently, Sedlacek III et al. (2018) have cautioned that rBC may be produced by laser-induced charring of organic substances in the SP2, which depends on the laser power. Such laser-induced charring could result in an overestimate of rBC. During our measurements, the laser power varied in the range 2.1-3.7 V, which is above the threshold to detect rBC with high efficiency (> 2V) (Sedlacek III et al., 2018). Though we cannot rule out an additional rBC contribution from**

charring of organic matter, it is likely this occurs in circumstances when the laser voltage is higher than that used in our study."

The following reference has been included in the revised manuscript.

Sedlacek, A. J., III, Onasch, T.B., Nichman, L., Lewis, E.R., Davidovits, P., Freedman, A., and Williams, L.: Formation of refractory black carbon by SP2-induced charring of organic aerosol, Aerosol Sci. and Technol., 52:12, 1345-1350, 2018, DOI:10.1080/02786826.2018.1531107.

*Also, it is not always appropriate to assume core-shell structure for coated BC due to the complex mixing state of BC, such as the case when BC is located off-center. Though the study did not utilize single particle off-line analysis to probe the complex internal mixing state of BC, it is useful to mention the effects from such BC structures on absorption and scattering signals (e.g., Sedlacek III, Arthur J., et al. "Determination of and evidence for non-core shell structure of particles containing black carbon using the Single Particle Soot Photometer (SP2)." Geophysical research letters 39.6 (2012)). The authors should discuss how such noncore- shell particles would affect their results.*

Agree. We have revised the section accordingly by adding the following:

Page 8, Line29.

"Sedlacek III et al. (2012) examined the structure of rBC containing particles using the 'lag time' technique and suggested that the core-shell model does not apply to all rBC –containing particles. A situation with non-core-shell structure (the case when BC is located off-center) arising due to the complex mixing state of BC may lead to uncertainty in determining the coating thickness of BC.  Our study assumes BC to be at the centre and a uniform coating around, in the absence of other measurements to understand the complex coating. A recent study by Liu et al., (2017) demonstrated good agreement between Mie-modelled scattering values using the core-shell approximation and the SP2-measured scattering cross-section for the BC with thicker coatings as is the case for the majority of particles in this study.  In addition, further  the particle scattering is relatively independent of particle morphology at the SP2 wavelength 1064nm (Moteki et al., 2010)."

The following references have been included in the revised manuscript.

Sedlacek, A. J., III,  Lewis, E. R., Kleinman, L., Xu,J., and Zhang, Q. : Determination of and evidence for noncore-shell structure of particles

containing black carbon using the Single-Particle Soot Photometer (SP2), Geophys. Res. Lett., 39, L06802, 2012, doi:10.1029/2012GL050905.

Moteki, N., Kondo, Y., and Nakamura, S.-i.: Method to measure refractive indices of small nonspherical particles: Application to black carbon particles, J. Aerosol Sci., 41, 513-521, 2010.

*For all instruments employed, also mention the model number in addition to the manufacturing company.*

**Complied with.**

*Be consistent in the use of units such as for flow rate and BC concentration. The use of different units for the same quantity makes it difficult to infer the values. Minimize the use of numeral values for comparison all the times. It may appear to the authors that these numerals are useful for comparison, but to me, it is a source of distraction. I recommend listing the values in Tables which the authors have already done. Authors can infer Table for values and focus on their findings.*

**Complied with for the entire manuscript.**

*General comments:*

*Page4 line 19: Add some references on the size of monomers for nascent BC.*

*Cite (2017): 166, Köylü, Ümit Özgür, et al. "Fractal and projected structure properties of soot aggregates." Combustion and Flame 100.4 (1995): 621-633;*

*Bhandari, Janarjan, et al. "Effect of thermodenuding on the structure of nascent flame soot aggregates." Atmosphere 8.9.*

**Complied with. We have added the following references in the revised manuscript.**

Köylü, Ü.Ö., Faeth, G.M., Farias, T.L., Carvalho, M.G.: Fractal and projected structure properties of soot aggregates, Combustion and Flame, 100, 621-633, 1995, ISSN 0010-2180, https://doi.org/10.1016/0010-2180(94)00147-K.

Bhandari, J., China, S., Onasch, T., Wolff, L., Lambe, A., Davidovits, P., Cross, E., Ahern, A., Olfert, J., Dubey, M., and Mazzoleni, C.: Effect of thermodenuding on the structure of nascent flame soot aggregates, Atmos. Meas. Tech. Discuss., https://doi.org/10.5194/amt-2016-270, 2016.

*Page 4 line19: Also, it is not correct to say that the coated BC is 'spherical' after coating. Rather I would prefer to use the term like 'compact' or 'collapsed' for coated BC core, though the core-shell model treats such coated BC as the spherical core for simplicity.*

**We agree and have revised the relevant portion (Page 4 , Line 17)**

**"However, it collapses to a compact BC particle with its cores being coated with other components via coagulation among aggregates and (or) via condensation of atmospheric vapours while aging in the atmosphere."**

*Page 4, line 27: What are you referring to using "these"? I am not clear.*

**This sentence is modified in the revised manuscript.**

**"All the aforementioned processes have implications for direct and indirect radiative forcing of BC".**

*Page4 line 32: What sources? Do you mean 'aerosol sources'? Be specific.*

**This is now modified to read 'aerosol sources'.**

*Page 5, line 24: Only one instrument is revealed. But the authors mention in the abstract that they used 2 instruments. What is the second instrument used? Specify the second instrument as well in the introduction section mentioning why the instrument was selected.*

**Complied with. The discussion about the ACSM is added here in the revised manuscript.**

*Page 5, line 25: Do the authors mean to say that the working of SP2 is based on long-term measurements only? Please rephrase the sentence.*

**The sentence is modified in the revised manuscript as below.**

**"To meet these objectives, state-of-the-art instruments were installed at Bhubaneswar for long-term measurements. These included a single particle soot photometer (SP2) for characterization of refractory BC (rBC) aerosols and an aerosol chemical speciation monitor (ACSM) for high-resolution chemistry of possible coating materials. The present study provides results from a yearlong database from a combination of these instruments, perhaps for the first time over the Indian region".**

*Page 6, line 12: Do you mean to say 'above ground level' by acronym AGL? Please mention the full name for the first time.*

**Yes. Complied with.**

*Page 6, line 28: By 'Supplementary Figure S1' are you referring for the figures in an appendix? If so rename the figure as A1.*

**Complied with. Supplementary figures are now renamed as Figure S1, S2 and so on.**

*Page 7, lines 1-8: As the numeral values for meteorological parameters are shown in Table 1, avoid using all these numerals in the text for comparison. Please minimize the use of numeral values in the text. Include only those specific values that are striking to discuss.*

Complied with. The sentence is modified in the revised manuscript as below.

"During SMS, the prevailing wind speed and temperature were moderate, and relative humidity (RH) was high (Table 1), while rainfall, associated with the monsoon, was extensive (total rainfall ~ 878 mm). Compared to the SMS, lower temperatures, winds and RH prevailed during the PoMS, with lower total rainfall (~ 201 mm). The lowest temperatures and RH of the year were seen during winter when calm wind conditions prevailed with almost no rainfall. The PMS witnessed the highest temperatures of the year (as high as 41 °C), moderately humid atmosphere and relatively higher wind speed compared to winter. During this season the region received a total rainfall of ~149 mm associated with thundershower events that led to high-velocity local winds. Details are given in Table 1"

*Page 7 lines 15-17: Include the model number and company name for each instrument*

Complied with. The sentence is modified in the revised manuscript as below.

"In the present study data was collected using a single-particle soot photometer (SP2) (Model: SP2-D; Droplet Measurement Technologies, Boulder, USA) and an Aerosol Chemical Speciation Monitor (ACSM) (Model: 140; Aerodyne Research Inc., USA)."

*Page8 line2: It is mentioned that the RI of 2.26 – 1.26i is used for BC. Is this RI representative for ambient BC aerosol in the region? I am aware of the use of the above-mentioned RI value for BC. However, the RI of 2.26 – 1.26i looks higher than usually used value for the RI of BC.*

The RI 2.26-1.26i has been widely used in the SP2 community to derive the scattering properties of rBC at the specified SP2 wavelength 1064nm (Moteki et al., 2010; Taylor et al., 2015; Laborde et al., 2012). The other RI, as commonly seen in the literature, are used for the other wavelengths, mainly for the optical properties in the visible range. We do not have a region-specific RI value for BC.

References:

Laborde, M., Schnaiter, M., Linke, C., Saathoff, H., Naumann, K. H., Möhler, O., Berlenz, S., Wagner, U., Taylor, J. W., Liu, D., Flynn, M., Allan, J. D., Coe, H., Heimerl, K., Dahlkötter, F., Weinzierl, B., Wollny, A. G., Zanatta, M., Cozic, J., Laj, P., Hitzenberger, R., Schwarz, J. P., and Gysel, M.: Single Particle Soot Photometer intercomparison at the AIDA chamber, Atmos. Meas. Tech., 5, 3077-3097, 10.5194/amt-5-3077-2012, 2012.

Liu, D., Whitehead, J., Alfarra, M. R., Reyes-Villegas, E., Spracklen, Dominick V., Reddington, Carly L., Kong, S., Williams, Paul I., Ting, Y.-C., Haslett, S., Taylor, Jonathan W., Flynn, Michael J., Morgan, William T., McFiggans, G., Coe, H., and Allan, James D.: Black-carbon absorption enhancement in the atmosphere determined by particle mixing state, Nat. Geosci., 10, 184-188, 10.1038/ngeo2901, 2017.

Moteki, N., Kondo, Y., and Nakamura, S.-i.: Method to measure refractive indices of small nonspherical particles: Application to black carbon particles, J. Aerosol Sci., 41, 513-521, 2010.

Taylor, J., Allan, J., Liu, D., Flynn, M., Weber, R., Zhang, X., Lefer, B., Grossberg, N., Flynn, J., and Coe, H.: Assessment of the sensitivity of core/shell parameters derived using the single-particle soot photometer to density and refractive index, Atmos. Meas. Tech., 8, 1701-1718, 2015.

*Page8 line 9: As mentioned earlier, be consistent in picking unit for a given quantity. Here for flow rate, you used cm3/min while in line 21 you used liters/minute.*

**Complied with.**

*Page 8, line 15: For the 40-100 nm range, mention clearly that the size is 'aerodynamic diameter'.*

**Complied with.**

*Page 8, line 26: The sub-heading 3.1 can be made more specific. By 'Mass and number concentration' only it is not clear what is being measured.*

**Complied with. The sub-heading 3.1 is modified as "BC mass and number concentrations" in the revised manuscript.**

*Pages 8-9 section 3.1: As mentioned earlier, use as least numeral values as possible in the discussion. All the numerals can be summarized in the table. Also, use the same units for particle concentration. In some cases, ng/m3 is used in some instance, µg/m3 is used for particle concentration.*

**Complied with. Particle concentration is expressed in µg m$^{-3}$ throughout the revised manuscript (relevant text is modified in abstract, discussion and conclusions). Units in figure 3(a) also are now modified.**

*Page 10, line 27: I am not clear about this '...reported for reported from...'. Please clarify this sentence.*

**Sorry; the repeated word is deleted.**

*Page 11, line 2: Is 'Figure S1' the same labelled as 'Figure A1' in the appendix?*

**In the revised manuscript, all the supplementary figures are labelled as Figure S1, S2, and so on. All these figures, corresponding text and labels are available as supplementary information in the revised manuscript.**

*Add label for each season in the map. In fig. A1 (a) and (b), it will be useful to mention the location of the distinct data points below south India as a note.*

**Complied with. Now labels are added in Figure S1.**

**"During SMS (and PoMS as well), a considerable amount of fire events are noticeable below south of India (over Srilankan region)"**

*Page 11, line 10: It is mentioned that the BC mass loading was lowest during PMS, but fire events were maximum during PMS throughout the Indian region, as shown in Fig. A (d). Is not it reasonable to expect a high concentration of BC?*

**The PMS is characterised by intense solar heating of the landmass and very little precipitation. As such, this season exhibits very high surface temperatures (going as high as 47-49 °C). Strong thermal convection resulting from intense solar heating of the dry land lifts the planetary boundary layer to higher altitudes, and with winds gaining speed, there is greater dispersion of the aerosols (Nair et al., 2007; Kompalli et al., 2014) leading to a substantial reduction in the surface concentrations. This stronger particle dispersion resulted in lower concentrations at the surface level, despite any increased contribution from fire events. Several studies have previously highlighted the presence of elevated aerosol layers over the Indian region during the pre-monsoon season (Satheesh et al., 2008; Babu et al., 2011).**

**Babu, S.S., Moorthy, K. K., Manchanda, R. K., Sinha, P. R., Satheesh, S. K., Vajja, D. P., Srinivasan, S., Kumar, V. H. A.: Free tropospheric black carbon aerosol measurements using high altitude balloon: Do BC layers build 'their own homes' up in the atmosphere?, Geophys. Res. Lett. 38, L08803, 2011, doi:10.1029/2011GL046654.**

**Satheesh, S. K., Moorthy, K.K., Babu, S.S., Vinoj, V., Dutt, C. B. S.: Climate implications of large warming by elevated aerosol over India. Geophys. Res. Lett. 35, L19809, 2008, doi:10.1029/2008GL034944.**

**Nair V. S., Moorthy K.K., Alappattu, D. P., Kunhikrishnan, P. K., George, S., Nair, P. R., Babu, S. S., Abish, B., Satheesh, S. K., Tripathi, S. N., Niranjan. K., Madhavan, B. L., Srikanth, V., Dutt, C.B.S., Badarinath, K. V. S., Reddy, R. R.: Wintertime aerosol characteristics over the Indo-Gangetic Plain (IGP):**

Impacts of local boundary layer processes and long-range transport; J. Geophys. Res. 112 D13205, 2007, doi: 10.1029/2006 JD008099.

*Page 14 lines7-9: The sentence "The figure reveals. . .." is not clear to me. Please rewrite the sentence*

**Complied with. The sentence is modified in the revised manuscript as below.**

**"In addition to the typical double-humped diurnal variation of BC mass concentration, which arises due to the combined effects of atmospheric boundary layer dynamics (Kompalli et al., 2014) and diurnal variation of the anthropogenic activities, very interesting links between BC core and relative coating thickness are noticeable from the Figure".**

*Page 14 line11, line 13: Do you mean to say that the RCT of BC is contributed by the day-night temperature difference in different seasons? I am not clear.*

**The sentence is modified in the revised manuscript as below for better clarity.**

**"and the amplitude of the BC variation has a marked seasonality. It is caused by the seasonal change in the diurnal variation of the ABL driven by seasonal changes in surface heating and resulting thermal convection. The highest amplitude occurs in winter since the diurnal variation of the ABL is greatest due to the high variation in surface temperature; with ΔT (i.e.$T_{max}$-$T_{min}$) ~ 12 °C over a 24 hour period. Conversely, the lowest amplitude occurs during the monsoon season, when thermal convection is highly suppressed due to the overcast sky, low surface heating and the surface energy balance being dominated by latent heat (the average diurnal amplitude of temperature variation, ΔT ~ 4.9 °C)."**

*Page 15 lines6-9: The sentence 'Interestingly. . .occurring' is a long sentence.*

**The sentence is split into two sentences in the revised manuscript as below.**

**"Interestingly, during the morning period when the BC mass concentration peaks due to the combined effect of the boundary layer dynamics (fumigation effect) and sources (rush hour traffic contribution), RCT was at a minimum. This suggests that fresh emissions from rush hour traffic, which would push up the BC concentration and lower the RCT, outweigh the fumigation effect; though both may be occurring around the same period."**

*Page 15, lines 13-16: The sentence 'Not. . .role' is again a long sentence and is not clear. Simplify your statement.*

**The sentence is split into simpler sentences as below:**

**"The diurnal variations in RCT are suppressed in the SMS and PMS compared to the winter and PoMS due to the seasonality of the boundary layer dynamics that modulates the concentrations of BC and the other condensing species. In addition to this, the wet scavenging by intense rains during the SMS ensures that a greater proportion of the remaining BC in the atmosphere is likely to be freshly emitted. Such extensive precipitation also leads to a reduction in concentrations of the coating substances. During the PMS, BC particles generally have larger core sizes, and the relative coating thickness is reduced in magnitude. These effects also play a role in shaping up the diurnal pattern."**

*Page 16, line 28: Chlorine is shown to be present in very low concentration even when air mass arrived from marine region to the sampling site during PMS. Is this concentration normal during the PMS as well?*

**Since the ACSM measures only NR-PM$_1$ and does not detect refractory materials (sublimation temperature > 600 ºC), which includes sea salt chloride, chloride measured by the ACSM is mainly from the sources other than of marine origin.**

*Page 17 lines 10-13: The sentence 'Further. . .. mixed' is not clear. In "absorption condensable species", do you mean 'absorption of condensable species' and the part "which already more internally mixed" is not clear.*

**The sentence is modified in the revised manuscript as below.**

**"The concentrations of freshly produced particles with little or no coating) arising from primary as well as secondary sources are, in general, greater during day. It enables more efficient adsorption of condensable species on these particles, compared to relatively aged particles during the night which are already coated or internally mixed due to aging. A greater fractional change can occur more quickly on fresh BC particles compared to particles which are already thickly coated since a much smaller amount of condensable material is required."**

*Technical comments:*

*Page 4 line 16: Correct the year for 'China et al., 2012' to make it '2013'.*

**Complied with.**

*Page 4 line 27: A sentence starts with a pronoun at the beginning of a paragraph. It makes it difficult to know what you are referring to using a pronoun in the very beginning of a paragraph.*

**The sentence is modified in the revised manuscript as below.**

**"All the aforementioned processes have implications for direct and indirect radiative forcing of BC."**

*Page 10 line3: What is represented by 'A' in equation (1)? Mention it.*

**Complied with.**

*Page 11 line1: add 'of' before 'larger-sized BC particles. . .'.*

**Complied with.**

*Page 20, line 2: Stronger is already a comparative adjective. Remove 'more' before stronger.*

**Complied with.**

---

## Author Comment (AC2) · 15 Dec 2019

**Response to reviewers**

**Observations of the reviewers are in italics, and our response is given below in bold letters.**

**Reviewer-2:** **ACP-2019-376-RC2**

*This paper describes measurements of black carbon (BC) containing particles and of non-refractory PM1 over 9-10 months at an urban site on the east coast of India. The data show some interesting seasonal trends and are worth publishing.*

**We thank the reviewer and appreciate the summary evaluation of the merit of our work.**

*However, the paper itself needs significant work. There are many missing words, misused words, confusing sentences and repetitive sections. It should not be such hard work to figure out what the authors are trying to say. Since the paper has several UK co-authors, I am surprised that it was not carefully edited by a native English speaker before submission. This paper needs major revisions before it is accepted for publication.*

**We are sorry. Now we have revised the manuscript thoroughly, considering all the suggestions of the reviewer, including those on the language.**

*Specific comments:*

*1- The introduction is rather repetitive and does not mention the second instrument, the ACSM. Please tighten up the language.*

**Complied with. Also commented by reviewer 1. We have removed the reduntant statements and also added the details of the second instrument. The following sentences are included in the revised manuscript.**

**Page5, Line 14: "Along with this, information on the condensable materials which act as coating substances and constantly alter the physiochemical properties of the BC containing particles, is also essential. Collocated mass spectroscopy-based high-resolution aerosol chemical composition measurements have been employed for this purpose (Liu et al., 2014; Gong et al., 2016)."**

**Page5, Line 25: "To meet these objectives, state-of-the the-art instruments were installed at Bhubaneswar, which included a single particle soot photometer (SP2) for characterization characterisation of refractory BC (rBC) aerosols and an Aerosol Chemical Speciation Monitor (ACSM) for high-resolution**

measurements of non-refractive submicron aerosol chemical composition for long-term measurements."

Page5: Line 31: "The contributions from distinct sources to BC concentrations and the association of coating on BC with possible condensable coating materials are examined, and the implications are discussed."

*2- page 2, lines 17 -25. It would be helpful to mention which months are included in each season.*

**Complied with.**

*3- page 2, lines 28 - 33: The statement "The diurnal pattern of sulphate resembled that of the RCT" and the statement "the coating on BC showed a negative association with sulphate" contradict each other. The association plot in Figure 11 is an interesting way to present the data, but the conclusions are tenuous and don't belong in the abstract. I would delete the sentence starting "Though the pre-monsoon: : :" and ending ": : :mixing state of BC."*

**Complied with. The sentence is deleted.**

**The following sentence is added in the revised manuscript (Page 2, Line29)**

**"Seasonally, the coating on BC showed a negative association with the mass concentration of sulphate during the pre-monsoon season and with organics during the post-monsoon season."**

*4- page 4, line 6: VOCs are not an aerosol species.*

**Agreed. We have rewritten the sentence.**

**"…phosphates, and secondary organic aerosols (SOA) originating from volatile organic compounds (VOC)".**

*5- page 4, line 19: A diameter of a few tens of nanometers is more likely for the primary spherules than for the chain agglomerates. Please cite a reference or correct the text.*

**Complied with. We have included the following references in the revised manuscript. (Page 4, Line 17)**

**References:**

**Bhandari, J., China, S., Onasch, T., Wolff, L., Lambe, A., Davidovits, P., Cross, E., Ahern, A., Olfert, J., Dubey, M., and Mazzoleni, C.: Effect of thermodenuding on the structure of nascent flame soot aggregates, Atmos. Meas. Tech. Discuss., https://doi.org/10.5194/amt-2016-270, 2016.**

**Bond, T. C., Doherty, S. J., Fahey, D. W., Forster, P. M., Berntsen, T., DeAngelo, B. J., Flanner, M. G., Ghan, S., Karcher, B., Koch, D., Kinne, S., Kondo, Y., Quinn, P. K., Sarofim, M. C., Schultz, M. G., Schulz, M., Venkataraman, C.,**

**Zhang, H., Zhang, S., Bellouin, N., Guttikunda, S. K., Hopke, P. K., Jacobson, M. Z., Kaiser, J. W., Klimont, Z., Lohmann, U., Schwarz, J. P., Shindell, D., Storelvmo, T., Warren, S. G., and Zender, C. S.: Bounding the role of black carbon in the climate system: A scientific assessment, J. Geophys. Res. Atmos., 118, 5380–5552, doi:10.1002/jgrd.50171, 2013.**

**Köylü, Ü.Ö., Faeth, G.M., Farias, T.L., Carvalho, M.G.: Fractal and projected structure properties of soot aggregates, Combustion and Flame, 100, 621-633, 1995, ISSN 0010-2180, https://doi.org/10.1016/0010-2180(94)00147-K.**

*6- page 4, line 27: Paragraph starts with the word "These," but it is not clear what "these" is referring to.*

**This sentence is modified in the revised manuscript as below:**

**"All the aforementioned processes have implications for direct and indirect radiative forcing of BC".**

*7- page 6, line 9: Figure 1b can be removed. No-one needs to see a picture of a shipping container.*

**Complied with. Revised figure 1 is shown below:**

[Figure]

**Figure 1: Geographic location of Bhubaneswar marked by a star symbol on the topographic map; the boundary of the Indo-Gangetic Plains (IGP) region is indicated with dotted lines.**

*8- page 6, line 11: Example of poor English usage "away from the proximity of." Just say "not near."*

**Complied with.**

*9- page 6, line 25: You refer to Figure S1, but the extra figures are in an appendix. Please label extra material consistently.*

**Complied with. Supplementary figures are renamed as Figure S1, S2 and so on in the revised manuscript.**

*10- page 6, line 30: I don't understand what the words in parentheses are conveying. If you mean that there are more fire events in those regions, please rephrase the sentence.*

**Complied with. The sentence is now rephrased and split into simpler sentences as below: (Page 6, Line 30)**

**"Figure S1 depicts the seasonal variation in the distribution of fires. The greatest number of fire events across the Indian region occur during the PMS. However, during other seasons, less intense fires are noticeable at the sub-regional scale, which are confined mostly to the northwest IGP during the PoMS, and to western, northeastern regions during winter".**

*11- page 7, lines 10-14: What about the 6-week gap in November-December? That is more than a brief gap and needs some explanation.*

**It has been explained in the revised manuscript as below:**

**"Only major gap in the data occurred during 11-November to 27-December, 2016, when measurements were paused due to logistical issues at the experimental site."**

*12- page 7, lines 17-18: Why mention instruments that are not relevant to this study?*

**Complied with. These sentences have been removed in the revised manuscript.**

*13- page 7, lines 31 – page 8, line 4: What do you mean by scattering enhancement? I think you are deriving the optical diameter of the coated particle from the scattering signal using a Mie Scattering model, but this description is garbled. Please phrase this more clearly.*

**Complied with. This discussion is rephrased in the revised manuscript as below:**

**"This signal is reconstructed using the leading edge only (LEO) fitting technique, which uses the leading edge of the unperturbed scattering signal before volatilization of the coating material becomes significant. This is used to reconstruct the full scattering signal (Liu et al., 2014). The reconstructed scattering signal and the BC core size ($D_c$) are used to derive the optical diameter of the BC particle or the coated BC size ($D_p$) by employing Mie calculations, where the whole particle is idealized as a two-component sphere with a concentric core-shell morphology".**

*14- page 8, line 15: Specify that this size range is vacuum aerodynamic diameter. The range is really more like 80 nm – 800 nm.*

**Complied with.**

*15- page 8, lines 19-20: It really doesn't seem necessary to discuss the pumps on the ACSM.*

**Complied with.**

*16 – page 10, lines 6-7 and elsewhere: Please pick one term, either counts or number and use it consistently throughout the paper including the figures.*

**Complied with. We have used 'number' instead of 'count' and count median diameter (CMD) is modified to number median diameter (NMD) throughout the manuscript, including figures.**

*17- page 10, first paragraph: Why is there a distinct jog at 40 nm in the number size distributions?*

**Thanks to the reviewer for this observation. In the present data analysis, the sum of the masses of all the single particle rBC detected formed the total rBC mass loading. A certain amount of rBC mass exists at core sizes, too small to be detected by the SP2, or too large, thus saturating the detector. In the present analysis, masses of such BC particles are predicted based on the extrapolation of a log-normal fit on the $D_c$ mass distribution (Liu et al., 2014). The values below 50 nm are obtained from such extrapolation and in the revised manuscript, the particles with $D_c$ < 50 nm are omitted, and the figure is modified (Supplementary figure S2).**

[Figure]

*18- page 16, line 16: What does "evolving least-squares fitting" mean?*

**This sentence has been modified in the revised manuscript as below:**

**"These size distributions were parameterized by least-squares fitting to an analytical monomodal log-normal distribution".**

*19- pages 10-11: I find this discussion extremely confusing and repetitive. The most important data is displayed in Figures 3 and 4 (and Table 4). I do not understand the point of averaging the size and number distributions over a season, taking the mode and getting a slightly different number than the average of the mode for individual data points. This does not add any new information and leads to repetitive discussion of the results. I would remove Figure 5, Table 2 and the associated discussion. Similarly, with the peak of the seasonal number size distributions – what new information does this give you beyond what you already know from the BC mass loading?*

**We partly agree. The idea of averaging over seasons is to provide inputs into models being developed in our own group and elsewhere, where the seasonal averages are needed. However, the repetitive discussions are avoided, and Figure 5 (and Table 2) is moved to supplementary section as supplementary figure S2 (and supplementary table-1) in the revised manuscript.**

*The discussion about previous work is split between page 10, lines 18-22 and page 11, line 29 – page 12, line 4. Please consolidate.*

**Complied with.**

*Finally, if you think you can tell the difference between local emissions during SMS with a smaller size and continental outflow with a larger size, why not make two entries in Table 3 for this study?*

**Complied with. These entries are made in the Table (Table-2 in the revised manuscript).**

*20- Pages 12-13: The discussion of RCT and ACT is confusing and repetitive. You are making a major assumption of core-shell morphology in order to calculate Dp and therefore RCT and ACT. Really all you can say is that you have a ratio that represents the amount of non-BC material associated with BC – you don't know the morphology of the particles nor how it changes with season. Morphology is likely to be quite different between fresh emissions during SMS and aged emissions during other seasons. I would not interpret RCT and ACT as literal diameters and coating thicknesses. In fact, I do not think ACT adds to the discussion. I would rewrite this section to present only the RCT data and include enough caveats that it is clear RCT is a representation, not an actual ratio of diameters.*

**We agree with the reviewer. It is correct that both RCT and ACT are used to represent the amount of non-BC material associated with BC and not an actual ratio of diameters. The information on the morphology of the BC, which may vary seasonally, is not available during our study period. The morphology would also be different for fresh and aged emissions. The coating thickness for individual particles is dependent on core sizes. However, the coating parameters estimated here are the bulk coating thicknesses in a given time window. It is calculated as the total volume of coated BC particles divided by the total volume of the rBC cores following Liu et al., (2014), which was used by subsequent studies (Liu et al., 2019, Brooks et al., 2019). As described by Liu et al., (2019), as the contribution from smaller particles to the integrated volume is very less, the bulk coating thickness values are generally independent of the uncertainties arising due to the presence of smaller particles. Some of the studies reported coating on rBC in terms of absolute coating thickness (ACT) in nm (Gong et al., 2016; Li et al., 2019; Cheng et al., 2018; Zanatta et al., 2019). Thus we used both these parameters for comparison of our values with the other regions. The RCT and ACT come from derived parameters that require Mie calculations based on a core-shell model that may not bear relation to reality. The caveat is that we assume the morphology of the particles ; they are**

spherical and coating is uniform (coated particle also is spherical). We agree that this is an over-simplification and the true morphology could be different. The RCT (and ACT) parameter provides a qualitative measures of the amount of condensed material that is present on the same particle as the rBC core.  We are using this to examine the extent of rBC mixing with other components in different seasons and compared to different regions. Further, using correlations with the bulk NR-PM1 composition,  we can obtain some insights into the coating material associated with rBC in different periods. Thus, we have used both the volume-weighted bulk RCT ($D_p/D_c$) and ACT (($D_p$-$D_c$)/2) in this study as representative diagnostics for the overall mixing state of the whole population of BC particles. We have modified the text in the revised manuscript to reflect the above discussion.

Page 8; Line 7

"These are calculated as the total volume of coated BC particles divided by the total volume of the rBC cores in a given time window (5 minutes) following Liu et al., (2014), which has been used by subsequent studies (Liu et al., 2019, Brooks et al., 2019a). It may be noted that the RCT and ACT used in this study come from derived parameters that require Mie calculations based on a core-shell model that may not bear relation to reality, and the RCT (and ACT) is not an actual ratio of diameters. The coating thickness for individual particles is dependent on core sizes. However, we have used the volume-weighted bulk RCT and ACT as representative diagnostics for the overall mixing state of the whole population of BC particles (Gong et al., 2016; Cheng et al., 2018; Liu et al., 2019). As described by Liu et al., (2019), since the contribution from smaller particles to the integrated volume is very less, the bulk coating thickness values are generally independent of the uncertainties arising due to the presence of smaller particles. Further, the information on the morphology of the BC, which would be different for fresh and aged emissions, is not available in this study.. The important caveat here is that we assume the morphology of the particles; they are spherical and coating is uniform (coated particle also is spherical). The RCT (and ACT) parameter provides a qualitative measure of the amount of condensed material that is present on the same particle as the rBC core.  We are using this to examine the extent of rBC mixing with other components in different seasons and compared to different regions. Further, using correlations

with the bulk NR-PM1.0 composition, we intend obtain some insights into the coating material associated with rBC in different periods".

References:

Brooks, J., Liu, D., Allan, J. D., Williams, P. I., Haywood, J., Highwood, E. J., Kompalli, S. K., Babu, S. S., Satheesh, S. K., Turner, A. G., and Coe, H.: Black carbon physical and optical properties across northern India during pre-monsoon and monsoon seasons, Atmos. Chem. Phys., 19, 13079–13096, https://doi.org/10.5194/acp-19-13079-2019, 2019.

Cheng, Y., Li, S.M., Gordon, M., and Liu, P.: Size distribution and coating thickness of black carbon from the Canadian oil sands operations. Atmos. Chem. Phys., 18, 2653–2667, 2018.

Li, K., Ye, X., Pang, H., Lu, X., Chen, H., Wang, X., Yang, X., Chen, J., and Chen, Y.: Temporal variations in the hygroscopicity and mixing state of black carbon aerosols in a polluted megacity area, Atmos. Chem. Phys., 18, 15201–15218, https://doi.org/10.5194/acp-18-15201-2018, 2018.

Liu, D., Joshi, R., Wang, J., Yu, C., Allan, J. D., Coe, H., Flynn, M. J., Xie, C., Lee, J., Squires, F., Kotthaus, S., Grimmond, S., Ge, X., Sun, Y., and Fu, P.: Contrasting physical properties of black carbon in urban Beijing between winter and summer, Atmos. Chem. Phys., 19, 6749–6769, https://doi.org/10.5194/acp-19-6749-2019, 2019.

Zanatta, M., Laj, P., Gysel, M., Baltensperger, U., Vratolis, S., Eleftheriadis, K., Kondo, Y., Dubuisson, P., Winiarek, V., Kazadzis, S., Tunved, P., and Jacobi, H.-W.: Effects of mixing state on optical and radiative properties of black carbon in the European Arctic, Atmos. Chem. Phys., 18, 14037–14057, https://doi.org/10.5194/acp-18-14037-2018, 2018.

*I also don't understand the point of Figure 7. You already have the information about the width of the distributions in Table 4. You mention multiple maxima, but have no interpretation. The discussion of Figure 7 on page 13 repeats the same information about sources and processing as on page 12, making this section very repetitive. I would remove Figure 7 and the associated discussion.*

We have complied with the reviewer's suggestion. Figure 7 has been moved to supplementary information in the revised manuscript (Supplementary Figure S3) and the repetitive discussion is removed.

*21- Page 12, lines12-14: "Both of these parameters: : :mixing state of the particles." I do not understand this sentence. Does "both" refer to RCT and ACT or to Dp and Dc? Either way, how can Dp not depend on the mixing state?*

**Yes. Both refer to RCT and ACT. We have modified the sentence in the revised manuscript accordingly.**

*22- Page 13, lines 1-2: "Intra-seasonal variability: : :Figure 6) is also higher during PoMS." I do not understand this sentence. Figure 6 shows daily values, not seasonal values. By eye, the variation in the daily points and the spread of the error bars looks very similar across PoMS, Winter and PMS. There are very few points during SMS, so it is hard to draw conclusions for that season.*

**The sentence is modified in the revised manuscript.**

**"Intra-seasonal variability (as highlighted by the wide range of frequency of occurrence of RCT and ACT values during the PoMS seen in the supplementary figure S3) is also higher during the PoMS".**

[Figure]

*23- Pages 15-17, Section 3.5: I have several questions about the ACSM data analysis. Why is ammonia so low? Was the aerosol not neutralized and do you have corroborating evidence? Or was the RIE_NH₄ incorrect for this instrument? Why not estimate OOA and HOA using the parameterization in Ng et al. (EST 2011)? This would give you additional information about local and regional sources.*

**We agree with the reviewer that the estimation of OOA and HOA provides information on the nature of sources. Detailed factorization of organics forms the scope for the future study, and is not attempted here. Periodic ionization efficiency calibrations were performed using ammonium nitrate, ammonium sulphate, and corresponding RIE_NH₄ values were updated in the DAQ of the ACSM. The reviewer made a good observation about the concentration of ammonium. We have estimated the aerosol neutralization ratio (ANR) (this information is not available in the present manuscript) in terms of the ACSM measured (m) $NH_4^+$ to predict (p) $NH_4^+$ ratio for different seasons and found a seasonal variability in the ANR values indicating ammonium deficit to fully neutralized aerosol system. A detailed analysis on this is being carried out. Earlier, Mahapatra et al., (2013) estimated chemical compostion of total suspended particulate (TSP) matter at Bhubaneswar using year-round filter-based sampling and have reported that both the acidic and basic components have significant seasonal variability. From the recent filter-based offline chemistry data there is evidence for seasonally varying ANR which indicated dominance of acidic ($NO_3^-$ and $SO_4^-$) over basic ($NH_4^+$, $Mg^{2+}$ and $Ca^{2+}$) atmospheres in different seasons at Bhubaneswar (unpublished data). One of the reasons for ammonium deficiency is possible heterogeneous reactions during the presence of a high number of pre-existing large particles and very high concentrations of acidic species (Pathak et al., 2009; Hsu et al., 2014). Collocated measurements of number size distributions of ultrafine and fine particles during the present study period at the site have also revealed the absence of new particle events due to high condensation sink (unpublished data) corroborating the above. The consolidated analysis of the aerosol chemistry from a combination of the size segregated off-line and online data methods is in progress to understand these aspects in detail.**

**References:**

Mahapatra, P.S., Ray, S., Das, N., Mohanty, A., Ramulu, T.S., Das, T., Chaudhury, G.R., Das, S. N.: Urban air-quality assessment and source apportionment studies for Bhubaneshwar, Odisha, Theor.Appl. Clim.,112,243-25, 2013.

Hsu, S.-C., C. S. L. Lee, C.A. Huh, R. Shaheen, F.-J. Lin, S. C. Liu, M.-C. Liang, Tao, J.: Ammonium deficiency caused by heterogeneous reactions during a super Asian dust episode, J. Geophys. Res. Atmos., 119, 2014, 6803–6817, doi:10.1002/2013JD021096.

Pathak, R.K., Wu, W.S., Wang, T.: Summertime PM2.5 ionic species in four major cities of China: nitrate formation in an ammonia-deficient atmosphere, Atmos. Chem. Phys., 9, 1711–1722, 2009. DOI: 10.5194/acp-9-1711-2009.

*24- Page 17, lines 1-7: "Even though: : :coating." I do not understand this sentence, partly because it is too long and convoluted, but also because the two parts contradict each other. You say in part a that concurrent peaks in RCT and sulphate suggest that sulphate is mixed with BC, but in part b, you say the opposite. You can't have it both ways. Or are you saying that the ACSM detects sulphate when it is mixed with BC, but not organic? That does not make sense.*

**This confusion has been cleared. We have rewritten this in the revised manuscript as below: (Page17, Line1)**

**"It is challenging to determine the exact coating material on the atmospheric BC particles in a multi-component system containing organic and inorganic aerosols, and gaseous vapours. The association between the diurnal variations of organics and sulphates and BC mixing state as represented by RCT presents two possibilities of having different coating material on BC during a day. Similar diurnal variations in RCT (as seen in Figure 6) and sulphate suggest the possibility of sulphate serving as the most probable material. However, organic matter can also contribute to the BC coating material due to its huge abundance in particles of submicron sizes. This is particularly true during the late evening periods, when concurrent peaks in the mass fraction of organics and rBC mass loading occur, a significant fraction of which could be secondary in nature. The extent of contribution of each species depends on processes such as gas-phase chemistry and production of condensable vapours and strength of the condensation sink".**

*25- Pages 17-18, Section3.7: Figure 11 is another way of comparing the diurnals for RCT and MF. While it is a nice visualization, I don't think it needs a new section repeating much of the discussion as in Section 3.6. I would combine the discussion in Sections 3.6 and 3.7.*

**Complied with. We have combined the sections 3.6 and 3.7, which describe the association between rBC relative coating thickness and NR-PM1 chemical species in diurnal and seasonal scales.**

*I also wonder if you have thought about the fraction of particles containing BC (i.e., BC number con./(BC num conc. + scatt num conc.))? This fraction is much higher in SMS than in PoMS or Winter and is lowest in PMS. The low value in PMS might be part of the reason that the association in Figure 11d and h is so poor since there is less overlap between the particle population detected with the ACSM and the population detected with the SP2.*

**Agreed with thanks. We have modified the text in the revised manuscript to include this possibility. (Page18, Line 28)**

**"It may be noted that it is difficult to decipher the exact coating on BC with the present approach, since the SP2 retrieves black carbon mass and provides a measure of co-existing material within the same particles (as measured by RCT) whereas the ACSM measures the mass of refractory material in the total submicron population. An examination of coating material can only be directly achieved by employing the instruments such as the soot particle aerosol mass spectrometer (Aerodyne SP-AMS) (Liu et al., 2018). However, the SP2 can determine both the rBC content of single particles and the optical size by light scattering for diameters between 200 and 400 nm. The coating thickness estimated within this range represents most of the particles which contribute significantly to the light extinction. A comparison of the proportion of rBC containing particles within the total population as a function of season sheds some light on interpreting variation throughout the year. In our study, the fraction of particles containing BC, i.e., the ratio of BC number concentration and total number concentration (BC number concentration + scattering number concentration) showed a clear seasonal variation. The fraction of BC containing particles was highest during the SMS (mean ~ 0.69 ± 0.11) and decreased through winter (~0.44 ± 0.16), PoMS (~0.36 ± 0.11) to reach the lowest value (~0.25 ± 0.10) during the PMS. This shows a gradual decrease in the overlap between the particle population detected with the ACSM and the population detected with the SP2 with changing seasons from SMS to PMS. This should be**

borne in mind while examining the association between the ACSM detected particle mass concentrations and the SP2 derived coating parameters."

Reference:

Liu, D., Taylor, J. W., Crosier, J., Marsden, N., Bower, K. N., Lloyd, G., Ryder, C. L., Brooke, J. K., Cotton, R., Marenco, F., Blyth, A., Cui, Z., Estelles, V., Gallagher, M., Coe, H., and Choularton, T. W.: Aircraft and ground measurements of dust aerosols over the west African coast in summer 2015 during ICE-D and AER-D, Atmos. Chem. Phys., 18, 3817–3838, https://doi.org/10.5194/acp-18-3817-2018, 2018.

*26- Figure 1: It is very hard to see the circle indicating the IGP in panel (a). I would delete panel (b). There is no need for a picture of a shipping container.*

**Complied with. Panel (b) has been deleted, and Figure 1 has been modified in the revised manuscript.**

[Figure]

**Figure 1:** Geographic location of Bhubaneswar marked by a star symbol on the topographic map; the boundary of the Indo-Gangetic Plains (IGP) region is indicated with dotted lines.

*27- Figure 2: The star symbol is not visible.*

**Figure 2 has been updated in the revised manuscript.**

[Figure]

28- *Figure 3: Indicate the seasons in panels (a) and (b).*

**Figure 3 has been updated in the revised manuscript as per the suggestion.**

[Figure]

**Figure 3: Temporal variation of daily mean (a) r$_{BC}$ mass concentration; and (b) number concentration of BC (bars) and non-BC scattering particles (filled circle). The vertical line passing through them is the standard deviation. The shaded portions demarcate the seasons.**

*29- Figure 4: Include the dashed lines in the legend. Panel (b) has circles not triangles. Please use either number or count, but not both.*

**Complied with. Figure 4 is modified in the revised manuscript.**

[Figure]

*Do you have any data covering the gap between end of May and August? Do the MMD and CMD really drop from PMS to SMS values over 6 weeks? Or could you have some kind of instrumental drift that causes both to increase over the displayed 10 months of data?*

Unfortunately, no data is available covering the gap between the end of May and August during the present study period due to technical issues with the SP2 optics (a drop in the SP2 laser power due to contamination of the optics owing to heavy particle loading). We rule out any instrumental drift as it has been periodically calibrated to account for any variation in the laser power and detector response. Notably, the present MMD values during the PMS are consistent with the values reported by Brooks et al., (2019) based on the aircraft experiments over the same region. They have reported that core MMD values which were 0.22 µm during the PMS (flights on 11–12 June 2018) dropped down to 0.20 µm with the onset of monsoon (flights on 30 June–11 July 2018; which were temporally just 2-4 weeks away from the pre-monsoon flights). They attributed it to change in the nature of air masses. The large scale changes in the air mass characteristics combined with the widespread precipitation across the Indian region associated with monsoon circulation contributes to changes in both the nature and strength of BC sources.

Reference:

Brooks, J., Liu, D., Allan, J. D., Williams, P. I., Haywood, J., Highwood, E. J., Kompalli, S. K., Babu, S. S., Satheesh, S. K., Turner, A. G., and Coe, H.: Black carbon physical and optical properties across northern India during pre-monsoon and monsoon seasons, Atmos. Chem. Phys., 19, 13079–13096, https://doi.org/10.5194/acp-19-13079-2019, 2019.

*30- Figure 11: "Speices" is mis-spelled in the x-axis label.*

It has been corrected in the revised manuscript.

*31- Table 1: "metrological" should be "meteorological"*

It has been corrected in the revised manuscript

*32- Table 3: "Shangai" is mis-spelled.*

It has been corrected in the revised manuscript

*33- Table 4: Are these averages of the daily values shown in the figures, or averages of all the underlying data? Please specify. Also, somewhere in the text you should state the time-base of the SP2 data.*

The values tabulated are averages of all the underlying data. Table-4 Table-3 in the revised manuscript) has been slightly modified as per the suggestion from reviewer-1. Time-base of the processed SP2 data is 5 minutes, and it is specified in the revised manuscript.

*34- Figure A1 (or S1?): Please decide if this section is an appendix or supplemental information.*

**In the revised manuscript, Figure S1, S2, S3 and S4 are available as supplementary figures, and the corresponding discussion is available as supplementary information.**

*Please label the panels with the season. It's not clear what the words in parentheses in the last sentence are supposed to mean.*

**Complied with. Now labels are added in Figure S1. Figure caption has been revised.**

[Figure]

**Figure S1:** Spatial distribution of Moderate Resolution Imaging Spectroradiometer (MODIS) fire radiative power (MODIS Thermal Anomalies / Fire locations Collection 6 product obtained from https://earthdata.nasa.gov/firms) for the representative months of different seasons; (a) August -2016 (SMS), (b) October -2016 (PoMS), (c) January -2017 (winter) and (d) May -2017 (PMS). A significant amount of fire events during PMS are seen over the Indian region. During the PoMS (fire events to confined to northwest IGP) and winter (fire events to confined to western, northeastern regions of India) less intense regional fire events are noticeable. During SMS (and PoMS as well), a considerable amount of fire events are noticeable below south of India (over Srilankan region).

---

## Author Response (AR2)

Interactive comments on **"Seasonal contrast in size distributions and mixing state of black carbon and its association with PM1.0 chemical composition from the eastern coast of India"**

5 **Response to Reviewer**

**Observations of the reviewers are in italics, and our response is given below in bold letters.**
*The revised paper is significantly improved, but still needs work as there are still many typos, poor wording choices, and some repetitive discussion. I have two major comments about paper content and some minor comments. This is not an exhaustive list of minor comments -- there are many more small*
10 *things that need correcting in the text.*

**We thank the reviewer and appreciate the summary evaluation of the merit of our work. Now we have revised the manuscript thoroughly, considering all the suggestions of the reviewer**
*This paper would still benefit from careful editing by a native English speaker.*

**The revised manuscript is thoroughly edited for improving the language by one of the co-authors**
15 **of the paper who is a native English speaker.**

**Major Comments:**

*1) I do not understand your distinction between the MMD and NMD data presented in Table 3 and in Table S1. If I understand correctly, the first is a fit to each individual 5 minute size distribution and then*
20 *taking the average for the season. The second is averaging the size distributions for a season, and then fitting. These two approaches should give the same result. In your case, the results are only slightly different and well within the error bars in Table 3. These differences are not meaningful and you should not spend paragraphs in the supplementary rehashing exactly the same seasonal analysis as in the main text. In your author response, you state that the fit to the seasonal average is useful as an input to models,*
25 *so that is all you need to say about it in the supplementary.*

**As suggested by the reviewer, we have now removed the supplementary figure S2, supplementary table S1 and corresponding discussion from the main (Page 11, Lines 7-11) and supplementary sections in the revised manuscript for better clarity.**

*2) However, I'm curious why you do get slightly different values for the NMD and MMD with the two*
30 *approaches. I wonder if it has something to do with the jog in the number size distribution at 50 nm from extrapolating the mass size distribution. You removed the data below 50 nm in Figure S2, but not in Figure 4a. In Figure 4a, you can see that the jog in the data pulls the log normal fit to smaller sizes. Did*

*you redo the fits to the size distributions in Figure S2 after removing the extrapolation? Is there a reason that you kept the extrapolation (and corresponding jog in number size distribution) in Figure 4a? I wonder if you would get the same results for the two approaches if you removed the extrapolation from the data for both.*

5     **In the present analysis, masses of BC particles that are too small or too large to be detected by the SP2 are predicted based on the extrapolation of a log-normal fit on the $D_c$ mass distribution following Liu et al., (2014). The values below 50 nm are obtained from such extrapolation, which are not used in the calculation of MMD or NMD. Figure 4 has been corrected as below in the revised manuscript (the particles with $D_c < 50$ nm are omitted).**

[Figure]

*3) Please put the tables in the order that you discuss them in the text. Table 3 (with the results from this measurement campaign) should be Table 2 and should be referred to on page 9 in the paragraph starting with line 26 where these results are discussed. I would make a table out of the comparison with mass loading from other campaigns on page 10, lines7-17. It is very hard to interpret a string of numbers in the text. This should be Table 3. The current Table 2 (with the comparison of mode diameters to results from other campaigns) should be Table 4.*

**Complied with. In the revised manuscript the tables are in the following order as per the reviewer's suggestion.**

**Table - 1:** Seasonal average of meteorological parameters, temperature (T), Relative humidity (RH), Pressure (P) and Wind speed (WS) and total accumulated rainfall. Maximum and minimum values (except for rainfall, which is seasonal total value) recorded in that season are also listed.

| Seasons | | T ( °C ) | P ( hPa ) | WS ( m s⁻¹) | RH ( % ) | Total accumulated rainfall (mm)* |
|---|---|---|---|---|---|---|
| **SMS** | Mean | $30.12 \pm 3.76$ | $995 \pm 3.6$ | $1.23 \pm 0.62$ | $77.1 \pm 17.5$ | 878 |
| | Minimum | 23.9 | 986.2 | 0 | 31 | |
| | Maximum | 35 | 1001.7 | 5.2 | 99 | |
| **PoMS** | Mean | $28.3 \pm 6.3$ | $1002 \pm 4.4$ | $1.07 \pm 0.67$ | $71.0 \pm 21.2$ | 201 |
| | Minimum | 21 | 993.9 | 0 | 17 | |
| | Maximum | 34 | 1010 | 4.8 | 99 | |
| **Winter** | Mean | $25.5 \pm 8.2$ | $1006 \pm 2.6$ | $0.95 \pm 0.53$ | $57.4 \pm 22.9$ | 2 |
| | Minimum | 18 | 998.1 | 0 | 8 | |
| | Maximum | 36 | 1001 | 3.4 | 99 | |
| **PMS** | Mean | $33 \pm 7.5$ | $999.8 \pm 3.6$ | $1.97 \pm 0.92$ | $67.1 \pm 28.2$ | 149 |
| | Minimum | 18.9 | 987.7 | 0 | 7 | |
| | Maximum | 41 | 1008.3 | 8 | 98.5 | |

*this value is total rainfall amount.

**Table 2**: A summary of the properties of the rBC concentrations, size distributions and its mixing state and scattering particle concentrations in different seasons: summer monsoon (SMS), post-monsoon (PoMS), winter and pre-monsoon (PMS). The values after ± symbol are standard deviations.

| Parameter | SMS | PoMS | Winter | PMS |
|---|---|---|---|---|
| BC mass concentration (µg m⁻³) | $1.22 \pm 1.03$ | $1.34 \pm 1.40$ | $1.94 \pm 1.58$ | $0.93 \pm 0.99$ |
| BC number concentration (cm⁻³) | $695 \pm 582$ | $583 \pm 616$ | $621 \pm 557$ | $218 \pm 239$ |
| Scattering particle concentration (cm⁻³) | $211 \pm 114$ | $690 \pm 471$ | $950 \pm 464$ | $548 \pm 349$ |
| Mass median diameter (µm) | $0.169 \pm 0.013$ | $0.182 \pm 0.012$ | $0.193 \pm 0.017$ | $0.219 \pm 0.011$ |
| number median diameter (µm) | $0.090 \pm 0.005$ | $0.100 \pm 0.006$ | $0.111 \pm 0.006$ | $0.128 \pm 0.010$ |
| Relative coating thickness | $1.16 \pm 0.04$ | $1.32 \pm 0.14$ | $1.34 \pm 0.12$ | $1.26 \pm 0.10$ |
| Absolute coating thickness (nm) | $24.24 \pm 9.9$ | $56.94 \pm 23.76$ | $65.01 \pm 15.80$ | $55.02 \pm 19.25$ |

**Table 3.** Average refractory black carbon (rBC) mass concentrations at selected locations

| Station and location | Observational period | mean $M_{BC}$ ($\mu$g m$^{-3}$) | Reference |
|---|---|---|---|
| Bhubaneswar, India | Jul. 2016- May 2017 | 1.34 ± 1.32 | Present study |
| Kanpur, India | Jan.-Feb. 2015 | 4.06 ± 2.46 | Thamban et al., (2017) |
| Gual Pahari, India | Apr.-May 2014 | 11 ± 11 | Raatikainen et al., (2017) |
| Mukteshwar, India | Feb.-Mar. 2014 | 1.0 ± 0.6 | Raatikainen et al., (2017) |
| London, England | Jan.–Feb. and Jul.–Aug. 2012 | 1.3 ± 1.1 | Liu et al., (2014) |
| Paris, France | Jan.–Feb. 2010 | 0.9 ± 0.7 | Laborde et al., (2013) |
| Beijing, China | Jan. 2013 | 5.5 | Wu et al., (2016) |
| Shanghai, China | Dec. 2013 | 3.2 | Gong et al., (2016) |
| Shenzhen, China | Jan. 2010 | 4.1 ± 3.8 | Huang et al., (2012) |
| Kaiping, China | Oct.-Nov. 2008 | 3.3 | Huang et al., (2011) |
| Lulang, China | Sep.-Oct. 2015 | 0.31 ± 0.35 | Wang et al., (2018) |

**Table 4.** A Summary of BC mass median diameter from a few selected studies representing different sources in distinct environments.

| S.No. | Location | Type of location | MSD mode/ MMD ($\mu$m) | Reference |
|---|---|---|---|---|
| **Urban/suburban locations** | | | | |
| *1.* | *Bhubaneswar, India* | *Urban/fresh urban emissions* | 0.169 ± 0.013 (July-Sept) | *Present study* |
| | | *Urban/continental outflow, aged BC* | 0.178-0.191 (Oct-Feb) | |
| | | Urban/with high solid fuel emissions | 0.219 ± 0.011 (Mar-May) | |
| 2. | Canadian oil sand mining (Aircraft studies), Canada | Urban/fresh urban emissions | 0.135-0.145 | Cheng et al. (2018) |
| 3. | Gual Pahari, India | Urban polluted/ fresh biofuel, crop residue | 0.221 ± 0.014 | Raatikainen et al. (2017) |
| 4. | Shanghai, China | Urban/pollution episode with high biomass burning | 0.230 | Gong et al. (2016) |
| 5. | Suzu, Japan | Urban/east Asian out flow site | 0.200 | Ueda et al. (2016) |
| 6. | An urban site in London, UK | Urban/traffic emissions | 0.119-0.124 | Liu et al. (2014) |
| 7. | Suburban site in Paris, France | Urban/traffic emissions | 0.100-0.140 | Laborde et al. (2013) |
| 8. | Sacramento, USA | Urban/fossil fuel emissions | ~0.145 | Cappa et al. (2012) |
| 9. | Tokyo, Japan | Urban outflow | 0.130-0.170 | Kondo et al. (2011) |
| 10. | Cranfield airport in UK | Aircraft emissions near source | 0.126 | McMeeking et al. (2010) |

| 11. | Regionally-averaged over flight segments over Europe | Near source to free troposphere | 0.170-0.210
(a) continental pollution (0.18–0.21);
(b) urban outflow (0.170±0.010) | McMeeking et al. (2010) |
|---|---|---|---|---|
| **Remote locations** | | | | |
| 12. | Lulang, Tibetan Plateau, China | High-altitude background | $0.160 \pm 0.023$ | Wang et al. (2018) |
| 13. | Mukteshwar, The Himalayas, India | High-altitude background / biofuel, crop residue outflow | $0.205 \pm 0.016$ | Raatikainen et al. (2017) |
| 14. | Northeastern Qinghai–Tibetan Plateau, China | Background site/biomass burning, aged BC | 0.187 | Wang et al. (2015) |
| 15. | Jungfraujoch, Switzerland | High-altitude background / biomass burning, aged BC | 0.220-0.240 | Liu et al. (2010) |

**Minor comments:**

*1) Page 5, lines 16 and 27. The "high-resolution aerosol chemical composition measurements" on line 16 is referring to high mass resolving power in the case of Liu et al. 2014 and high time-resolution in the case of single particle SPAMS measurements in Gong et al., 2016. The ACSM has neither high mass resolving power nor particularly high time-resolution. Please do not use the same phrase to refer to ACSM measurements in line 27. In addition, it's "non-refractory" not "non-refractive."*

**Complied with. The sentence in line 15 has been removed in the revised manuscript. The phrase 'high-resolution' is removed in line 27. Also "non-refractive" is corrected as "non-refractory".**

*2) Page 6, lines 27-29. Phrase starting "In supplementary Figure S1…" and ending "…(PMS)." is not a complete sentence. Maybe combine it with the following sentence.*

**Complied with. The sentences have been combined in the revised manuscript.**

*3) Page 7, lines 4-10. Please put the rainfall amounts in Table 1 in order to avoid a lot of numbers in the text. The last sentence of this paragraph is not necessary since the authors already pointed the reader to Table 1 earlier in the paragraph.*

**Complied with. We have included the total rainfall amounts in Table-1. Also the last sentence (line10) has been removed in the revised manuscript.**

*4) Page 9, line 6: Start a new paragraph when you change subjects from SP2 to ACSM.*

**Complied with.**

*5) Page 9, lines 26-27. How can a range of 0.31 to 0.26 give a mean of 1.23? Please include the correct numbers.*

**Sorry for the inadvertent mistake. The sentence is corrected in the revised manuscript.**

5    **"During SMS the daily mean mass concentration ranged between 0.31-2.6 µg m$^{-3}$ with a seasonal mean ~1.23 ± 1.03 µg m$^{-3}$"**

*6) Page 12, line 19. Should be Figure 5, not Figure 6.*

**Figure is corrected now as "Figure 5"**

*7) Page 18, line35. Liu et al. 2018 is not a reference for the SP-AMS.*

10    **Thanks to the reviewer for this observation. The reference is corrected as "Gong et al., (2016)" .**

*8) Page 44, Table 2. Why do you have the results from Table S1 in this table? You should use the numbers from Table 3 – these are your results for this campaign.*

**We agree with reviewer's suggestion. The values are now modified (Table-4 in the revised manuscript).**

15    *9) Supplementary page 5. The M_0 units in Table S1 are ug/m3, but the numbers are actually ng/m3 according to the text. Please convert the numbers to ug/m3 in the text and the table.*

**Supplementary Table S1 has been removed in the revised manuscript.**

[revised manuscript text omitted]